# Flexible and broadband colloidal quantum dots photodiode array for pixel-level X-ray to near-infrared image fusion

Jing Liu[1,2,3], Peilin Liu[1], Tailong Shi[1], Mo Ke[1], Kao Xiong[1], Yuxuan Liu[1], Long Chen[1], Linxiang Zhang[1], Xinyi Liang[1], Hao Li[1], Shuaicheng Lu[1,3], Xinzheng Lan[1,2], Guangda Niu [1,2], Jianbing Zhang [1,2,3], Peng Fei [1,2], Liang Gao [1,2,3] ✉ & Jiang Tang [1,2,4] ✉

Combining information from multispectral images into a fused image is informative and beneficial for human or machine perception. Currently, multiple photodetectors with different response bands are used, which require complicated algorithms and systems to solve the pixel and position mismatch problem. An ideal solution would be pixel-level multispectral image fusion, which involves multispectral image using the same photodetector and circumventing the mismatch problem. Here we presented the potential of pixel-level multispectral image fusion utilizing colloidal quantum dots photodiode array, with a broadband response range from X-ray to near infrared and excellent tolerance for bending and X-ray irradiation. The colloidal quantum dots photodiode array showed a specific detectivity exceeding $10^{12}$ Jones in visible and near infrared range and a favorable volume sensitivity of approximately $2 \times 10^5$ $\mu C$ $Gy^{-1}$ $cm^{-3}$ for X-ray irradiation. To showcase the advantages of pixel-level multispectral image fusion, we imaged a capsule enfolding an iron wire and soft plastic, successfully revealing internal information through an X-ray to near infrared fused image.

Multi-spectral image fusion is a technique that extracts the most pertinent information from different-wavelength source images into a unified image, with the goal of providing richer and more valuable information for subsequent applications, such as machine vision[1], autonomous vehicles[2], medical diagnosis[3] and other artificial intelligences[4]. Existing approaches for multi-spectral image fusion typically rely on vision algorithms, including multi-scale transformation[5], deep learning[6] and etc., at the sacrifice of resolution mismatch, overloaded computing resources and complicated systems[7]. With the advancement of photodetectors that have broader response range, pixel-level image fusion can be a more practical approach, where multi-spectral images are captured using just one photodetector. This approach simplifies imaging processes and systems, with the additional benefits of conserving computational resources and reducing energy consumption. For example, traditional InGaAs photodetectors have been modified to broaden their response range from 0.9 to 1.7 μm to 0.4–1.7 μm for visible-infrared pixel-level image fusion[8], yielding more informative images in the inclement weather.

Pixel-level multi-spectral image fusion (PLMSIF) of X-ray, visible and infrared is highly desired in various areas such as medical imaging[9], security monitoring[10] and nondestructive testing[11]. As for application in medical imaging, the X-ray image emphasizes the inorganic skeleton texture, while the visible image supports the

[1]Wuhan National Laboratory for Optoelectronics and School of Optical and Electronic Information, Huazhong University of Science and Technology, 1037 Luoyu Road, 430074 Wuhan, P. R. China. [2]Optical Valley Laboratory, 430074 Wuhan, P. R. China. [3]Wenzhou Advanced Manufacturing Technology Research Institute of Huazhong University of Science and Technology, 225 Chaoyang New Street, 325105 Wenzhou, P. R. China. [4]National Engineering Research Center for Laser Processing, 1037 Luoyu Road, 430074 Wuhan, P. R. China. ✉e-mail: highlight@hust.edu.cn; jtang@mail.hust.edu.cn

assessment of appearance, and the infrared image provides a detailed description of organic tissue structure. Combining X-ray, visible and infrared images into one single image can effectively and comprehensively construct the complete medical atlas, as realized by the traditional approach (Fig. 1a) using three individual photodetectors for X-ray, visible, infrared and then applying a vision algorithm. This system requires complex vision algorithms and extensive computing resources to compensate for the differences in pixel position and resolution between the three types of photodetectors, impeding the development of artificial intelligence in medical imaging. As another increasingly active demand for comfortable and real-time medical imaging, wearable and flexible photodetectors also need to be taken into consideration and developed to fit irregular biology surface and

improve comfort level. However, as far as we are concerned, there is no report on one single flexible photodetector capable of capturing X-ray, visible and infrared images to achieve image fusion (Fig. 1b). This approach could be useful for flexible lens-free imaging, such as biomedical measurement and medical diagnosis[12].

Various materials such as halide perovskites[13,14], organic semiconductors[12], two-dimensional materials[15,16] and colloidal quantum dots (CQDs)[17,18] have emerged, enabling flexible and wide detection range beyond traditional silicon and InGaAs photodetectors. Halide perovskites are ultra-sensitive and have a low detection limit for X-ray and visible detection due to their high absorption coefficient and high μτ product, but they show poor performance for infrared detection owing to their large bandgap[19,20]. Organic semiconductors

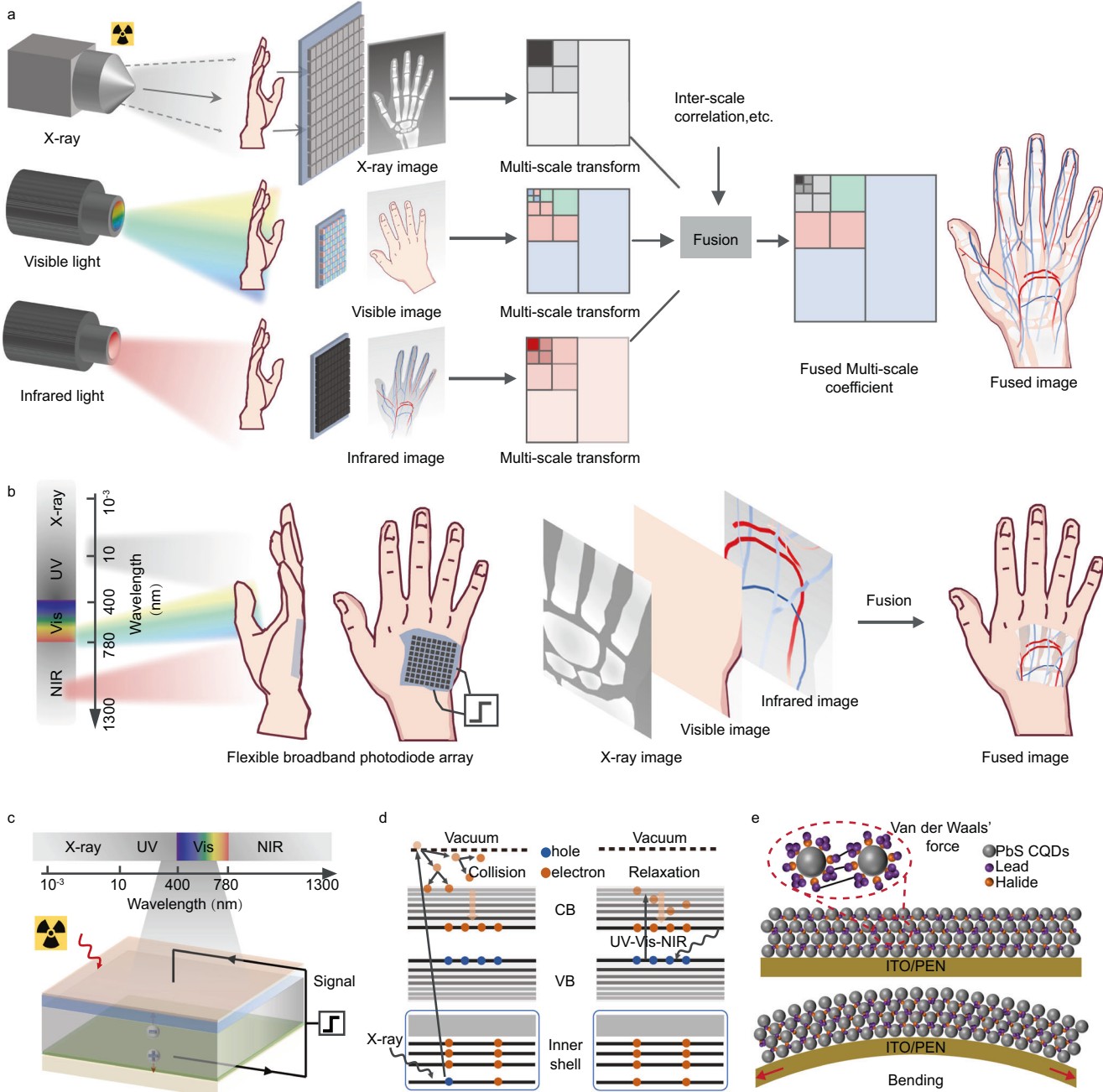

**Fig. 1 | Pixel-level X-ray to infrared image fusion and work mechanism.** **a** Traditional pixel-level X-ray to infrared image fusion by using three photodetectors at corresponding wavebands and vision algorithm. **b** Pixel-level X-ray to infrared image fusion by using one single flexible and broadband photodetector. **c** Working principle of broadband PbS CQDs photodetector from X-ray to infrared. **d** Non-equilibrium carriers generation of PbS CQDs excited by X-ray and ultraviolet-visible-infrared irradiation. **e** Schematic diagram of PbS CQDs film under bending state.

have achieved ultra-low dark current, large linear dynamic range and excellent flexibility but with limited response range and poor X-ray irradiation resistance[21]. Two-dimensional materials such as graphene exhibit fast photoresponse and ultra-broadband response from visible to terahertz, but they are too thin to efficiently absorb X-ray and have limited capacities for imaging array[22]. PbS CQDs are widely recognized for their excellent visible and infrared photodetection capabilities, which are attributed to their tunable bandgap, high absorption coefficient and low-temperature solution processing[23–25]. Actually, these materials contain heavy element Pb which is a strong absorber for X-ray because the absorption coefficient of X-ray is proportional to the fourth power of atomic number (Pb, 82). Furthermore, as shown in our manuscript, PbS CQDs exhibit much better X-ray robustness compared to their bulk counterpart. Hence, PbS CQDs are at least one of the best choices for pixel-level X-ray to infrared image fusion.

The operating principle of PbS CQDs broadband photodetector is illustrated in Fig. 1c, in which PbS CQDs film transfers energy of X-ray and ultraviolet-visible-near infrared (UV-Vis-NIR) photons to electron-hole pairs due to photoelectric effect and Compton scattering as shown in Supplementary Fig. S1. In the photoelectric effect process, an X-ray photon is mainly absorbed by Pb atom and an inner electron is ejected and re-absorbed by the solid. During the Compton scattering process, the X-ray photon changes its path and speed to transfer energy to the electron and a recoil electron is generated (Supplementary Fig. S1). As shown in Fig. 1d, for the X-ray detector, the ejected shell electrons with high kinetic energy release many electrons by collision ionization and relax down to the conduction band minimum. And the holes are left in the valance band maximum to create electron-hole pairs. Under UV-Vis-NIR light illumination, the electrons near the valence band maximum are excited into the conduction band by absorbing the photon energy and the holes are preserved to generate carriers. Van der Waals interaction between adjacent dots allows the slipping of CQDs without broken bonds and new defects under the bending state (Fig. 1e), which supports the desirable flexibility of CQDs devices (Supplementary Fig. S2).

In this work, we demonstrated a flexible PbS CQDs photodiode array with ultra-broadband response range from X-ray to near infrared (NIR) which were fabricated by the low-temperature method. The structure and thickness of our photodiodes were well designed to reduce carrier recombination via interface defects and improve carrier generation in the depletion region. The pixel photodiode obtained impressive performance with a low dark current density (50.9 nA cm$^{-2}$ at −1 V), a large linear dynamic range (>85 dB) and a high detectivity (1.01 × 10$^{12}$ Jones) under Vis-NIR illumination. It could be operated at a very low voltage (0.1–1.25 V) with an area sensitivity of 17.8 μC Gy$^{-1}$ cm$^{-2}$ which is comparable to the commercial α-Se detector under X-ray irradiation, and sustained high performance after bending repeatedly or long-time X-ray irradiation. We further showcased the nondestructive testing of a capsule using our PbS CQDs photodiode array, which demonstrated great potential for pixel-level X-ray to infrared image fusion by one single photodetector.

## Results and discussion

The as-prepared flexible 100 × 100 PbS CQDs photodiode array in the inset of Fig. 2a shows 20 × 20 mm$^2$ active area with 0.9 × 0.9 mm$^2$ pixel area and 0.1 mm pixel pitch patterned by a shadow mask. Figure 2a demonstrates the structure of a single photodiode with polystyrene naphthalate (PEN) as flexible substrate and a PIN heterojunction. The cross-section scanning electron microscope (SEM) image of the photodiode in Fig. 2b presents well-bedded stacks of ITO/NiO$_x$/PbS CQDs/C$_{60}$/ZnO/Al. The bottom electrode ITO, p-type transport layer NiO$_x$ and n-type transport layer ZnO were sequentially deposited by magnetron sputtering on the flexible substrate. The transmittance spectra in Supplementary Fig. S3 demonstrate the high transmittance (>80%) of the transport layers (NiO$_x$ and ZnO) and flexible conductive

substrate (ITO/PI) from 400 to 1300 nm. The interface layer C$_{60}$ and patterned top electrode Al were deposited by thermal evaporation. The vacuum processes are beneficial for reproducibility and stability.

The optimum thickness of the PbS CQDs layer is determined by the diffusion and drift length of photogenerated carriers[26–28] and adequate X-ray absorption. The active layer of PbS CQDs was fabricated by spin-coating with a thickness of ~900 nm enabling ~5% X-ray absorption (Supplementary Figs. S5, S11). PbS CQDs were synthesized by cation-exchange method[29] with a tight size distribution around 2.8 nm and good crystallinity as shown in Supplementary Fig. S4a. The solution-phase ligand exchange method[30] was used to exchange long-chain organic ligand oleic acid (OA) by shorter inorganic ligands PbX$_2$ (X = Br, I). The organic ligands were removed sufficiently during the ligand exchange process, certified by no characteristic peaks (C−H and COO$^-$) of OA in Fourier transform infrared (FTIR) spectra (Supplementary Fig. S4b). The PbX$_2$ capped CQDs show obvious exciton absorption peak in solution and film state (Supplementary Fig. S4c), illustrating good monodispersity of CQDs. The CQDs film shows a little redshift from 940 nm to 1040 nm compared to solution state due to increased inter-dot coupling[30]. The high-quality CQDs film was also investigated by atomic force microscope (AFM), X-ray diffraction (XRD) and X-ray photoelectron spectroscopy (XPS) as shown in Supplementary Fig. S4d−f. As a result, the CQDs film is smooth with a very low roughness of ~5 nm. The diffraction peaks agree well with the standard PDF card of PbS, demonstrating that there was no excessive PbI$_2$ in the CQDs film. There is only one type of peaks for Pb−S bond in the XPS Pb 4f spectrum with the binding energies of 138 eV and 142 eV, proving that the CQDs film is free of oxidation. The energy band alignment of PbS CQDs photodiode in Fig. 2c promotes efficient extraction of photo-generated electrons and holes and reduces recombination at electrodes.

The current density–voltage (J−V) curves of our flexible PbS CQDs photodiode are shown in Fig. 2d. The device exhibits a dark current density as 50.9 nA/cm$^2$ at −1 V bias and a rectification ratio of around 1000 at ±1 V bias, where the bandgap of our PbS CQDs is 1.18 eV. The photocurrent increases and saturates at a low reverse bias under the excitation of 970 nm LED light source. The photogenerated carriers can be completely collected under increased reverse bias due to a wider depletion region and stronger built-in field. Figure 2e demonstrates the device photocurrent under 970 nm illumination with the light density ranging from 7 × 10$^{-5}$ to 2 mW cm$^{-2}$. The linear dynamic range (LDR) is calculated as more than 85 dB at −2 V bias. The LDR of the photodiode under 390 nm, 530 nm and 780 nm illumination at −2 V bias were further measured, as shown in Supplementary Fig. S6. The photocurrent of the device increases linearly within the test range, which demonstrates wide LDR over a broad spectral range.

The external quantum efficiency (EQE) and responsivity (R) at different wavelengths at −2 V bias are shown in Fig. 2f, consistent with the absorption of the PbS CQD film in Supplementary Fig. S4c. The optimum EQE and R are 76.6% and 0.38 A W$^{-1}$, respectively, at the wavelength of 620 nm. The flexible PbS CQD photodiode also shows a high responsivity under infrared radiation. The EQE and R are as high as 43.4% and 0.34 A W$^{-1}$ at 970 nm. However, the EQE and R are significantly reduced at the wavelength less than 400 nm due to the intense absorption of the flexible substrate (Supplementary Fig. S3c).

Figure 2g exhibits the noise current spectral density with the frequency from 1 Hz to 40 kHz at −1 V bias. The noise current, mainly composed of generation-recombination (G-R) noise, 1/f noise and shot noise, was measured by a lock-in amplifier and decreases with the frequency. In the low-frequency region (<1 kHz), the noise current is dominated by 1/f noise due to bulk or surface traps of the PbS CQDs layer causing carrier scattering during transport. The noise current is mainly originated from G-R noise in the frequency range from 1 to 40 kHz. And the shot noise is independent on the frequency, which demonstrates the lower limit of the noise. The total calculated noise

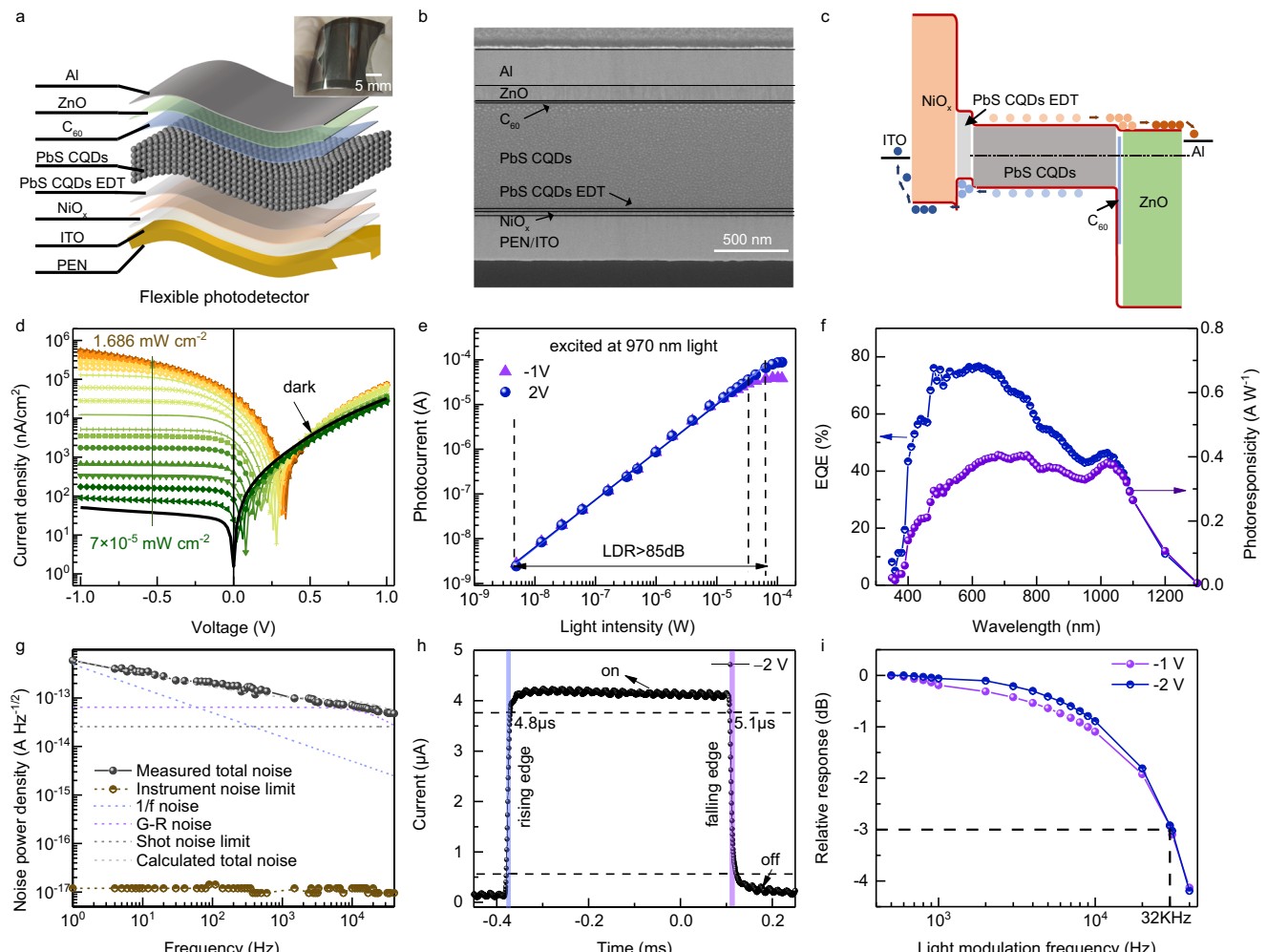

**Fig. 2 | Device performance of PbS CQDs photodiode under UV-Vis-NIR light illumination. a** Schematic diagram of flexible PbS CQDs photodiode. The inset is a photograph of flexible PbS CQDs photodiode array with a scale bar of 5 mm. **b** Cross-section SEM image of flexible PbS CQDs photodiode. **c** Energy band diagram of flexible PbS CQDs photodiode. **d** Current density–voltage curves under dark and 970 nm LED illumination with different power density. **e** Linear dynamic range at bias of −1 V and −2 V. **f** Broadband EQE and responsibility spectra at −2 V bias. **g** Measured and calculated noise current spectral density. The calculated limits of 1/$f$ noise, generation-recombination (G-R) noise and shot noise are included. **h** Temporal response at −2 V bias illuminated by 780 nm LED with the power density of 0.1 mW cm⁻². **i** −3 dB bandwidth at bias of −1 V and −2 V.

(grayish dotted line) fit well with the measured noise (black solid line). The specific detectivity ($D^*$) and noise equivalent power (NEP) were calculated by the measured noise current ($I_n$) and responsivity ($R$) using the equations in the Methods section. The NEP of our flexible PbS CQDs photodiode at 1 Hz is $1.87 \times 10^{-13}$ W at zero bias (self-powered mode) and $1.74 \times 10^{-12}$ W at −1 V bias as shown in Supplementary Fig. S7a, d. The measured $D^*$ is, respectively, $1.01 \times 10^{12}$ Jones at zero bias and $1.52 \times 10^{11}$ Jones at −1 V bias at 1100 nm. And wavelength-dependent $D^*$ is further demonstrated in Supplementary Fig. S7b–f, which indicates sensitive and broadband response of our flexible PbS CQDs photodiode.

In addition, the temporal photoresponse of our flexible PbS CQDs photodiode is shown in Fig. 2h measured at −2 V bias under 780 nm LED illumination with the power density of 0.1 mW cm⁻². The rise time and fall time are, respectively, 4.8 and 5.1 μs, defined as time interval between 90% and 10% maximum photocurrent. And the rise time and fall time at −1 V bias are similarly 4.9 and 5.2 μs (Supplementary Fig. S8) due to depleted junction. Furthermore, the frequency response in Fig. 2i shows our flexible PbS CQDs photodiode has a −3 dB bandwidth about 32 kHz at −1 and −2 V bias, corresponding to the response time. Our photodiode remains the same photoresponse and a little decreased dark current when stored in dry

air for 720 h without encapsulation (Supplementary Fig. S9), demonstrating good stability. And our devices exhibit good uniformity with the standard deviation of dark current and EQE of 1.4 nA and 1.8%, respectively (Supplementary Fig. S10). Supplementary Table S1 summarizes the performance of previously reported flexible photodetectors. Our device simultaneously has high $D^*$, large LDR and fast response rate, which demonstrates the state-of-the-art comprehensive performance.

Flexible X-ray detectors are very useful for dental checking as they enable better image quality and reduce X-ray exposure. As expected, our flexible PbS CQDs photodiode exhibits sensitive X-ray detection, as shown in Fig. 3. The absorption coefficient of PbS is shown in Fig. 3a based on the ESTAR/XCOM database[31], which is higher than some typical semiconductors such as Si and α-Se on account of its large average atomic number. The absorption capability is generally proportional to the 4th power law of the atomic number, Pb has a large atomic number of 82 and hence PbS enjoys strong X-ray absorption. Supplementary Fig. S11a demonstrates the mass absorption efficiency of some semiconductors to 50 keV X-ray photons. Bulk PbS and PbS CQDs (Supplementary Table S2) with higher absorption coefficient than traditional Si and α-Se allow thinner film to achieve adequate X-ray absorption.

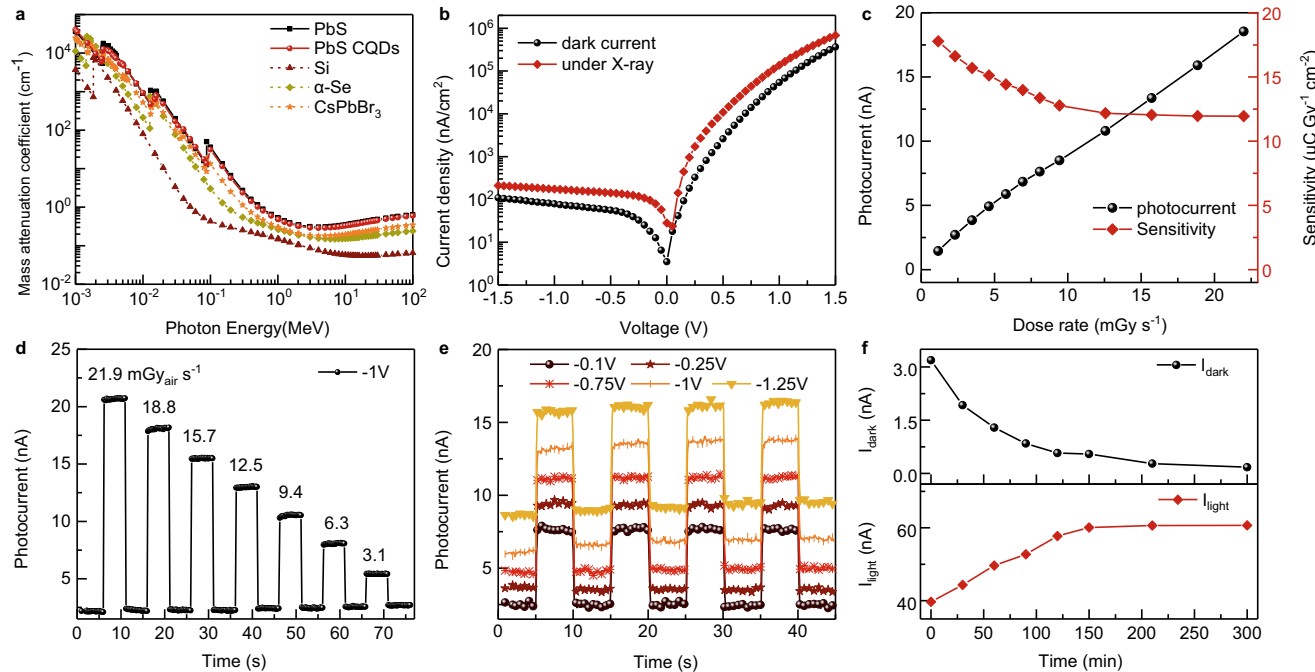

**Fig. 3 | Performance of flexible PbS CQDs photodiode under X-ray irradiation.**
**a** Mass absorption coefficient of bulk PbS, PbS CQDs, Si, α-Se and CsPbBr₃ as a function of photon energy. **b** Current density–voltage curves of the PbS CQDs photodiode under dark and X-ray irradiation with 8.1 mGy$_{air}$ s$^{-1}$ dose rates. **c** Photocurrent and sensitivity to X-ray varying with the dose rate at −1 V bias. **d** Transient response to X-ray under different dose rates at −1 V bias. **e** Transient response at various biases under 8.1 mGy$_{air}$ s$^{-1}$ dose rates. **f** X-ray irradiation stability of PbS CQDs film.

We measured the performance of our flexible PbS CQDs photodiode by a tungsten anode X-ray tube with the average photon energy of 30 keV. The current density–voltage (J–V) curves of our photodiode are shown in Fig. 3b under 8.1 mGy$_{air}$ s$^{-1}$ dose rates X-ray irradiation. The photocurrent nearly remains constant in a bias range from −0.1 to −1.5 V as bias-independent X-ray response, indicating the X-ray generated carriers can be completely collected by built-in electric field and low bias. The dark current density is low, about 12.6 nA/cm² at −0.1 V bias. It suggests great potential of our device for a flexible imaging array, which requires low working bias and bias-independent X-ray response. The temporal response under X-ray irradiation with various dose rates from 3.1 to 21.9 mGy$_{air}$ s$^{-1}$ is exhibited in Fig. 3d. The photocurrent sensitivity reduces along with the decreased dose rate of X-ray irradiation.

The sensitivity of our device was further calculated by photocurrent and X-ray dose rate using the equation in the Methods section. Figure 3c demonstrates the evolution of photocurrent and sensitivity under different dose rates. The photocurrent increases linearly with the dose rate and the sensitivity is almost constant at a high dose rate range from 8.1 to 21.9 mGy$_{air}$ s$^{-1}$. The sensitivity slightly improves from 11.9 to 17.8 μC Gy$^{-1}$ cm$^{-2}$ when decreasing the dose rate to a low range (8.1 mGy$_{air}$ s$^{-1}$) because of the larger response gain from shallower traps at low irradiance[32]. Despite its thin thickness (900 nm), our detector reached a sensitivity of 17.8 μC Gy$^{-1}$ cm$^{-2}$, which is comparable to the commercial α-Se detector (1–20 μC Gy$^{-1}$ cm$^{-2}$)[33]. It should be noted that the volume sensitivity of the device is about 2 × 10⁵ μC Gy$^{-1}$ cm$^{-3}$ at the lowest bias voltage of 0–0.1 V, which is comparable with that of the reported flexible X-ray direct detectors using new materials (Supplementary Table S3)[34]. The temporal response of our flexible PbS CQDs photodiode at different biases under 8.1 mGy$_{air}$ s$^{-1}$ X-ray irradiation is shown in Fig. 3e. As expected, larger bias enable stronger photoresonse at the expense of deteriorated dark current density. After 8.1 mGy$_{air}$ s$^{-1}$ dose rates irradiation for 27 h, our flexible PbS CQDs photodiode demonstrates only ~0.16 nA increase of dark current and almost no change of

photocurrent (Supplementary Fig. S12), demonstrating good stability under X-ray irradiation.

Surprisingly, our PbS CQDs film demonstrates excellent stability to X-ray irradiation, as shown in Fig. 3f. After 5.5 mGy$_{air}$ s$^{-1}$ dose rates X-ray irradiation for ~200 min, the dark current of Au/PbS CQDs/Au photoconductor is reduced by 6 times and the photocurrent is increased by 1.5 times. The photoresponse of 7 PbS CQDs films remains stable after X-ray irradiation (5.5 mGy$_{air}$ s$^{-1}$) for one week with a total dosage of 3326 Gy$_{air}$ dosage (Supplementary Fig. S13). Apparently, X-ray irradiation is beneficial for CQD device performance. However, we did not observe similar behavior on PbS polycrystalline film prepared by chemical bath deposition (CBD) (Supplementary Fig. S11b). After the same dosage of X-ray irradiation, the dark current of Au/CBD PbS/Au photoconductor is increased by ~2 times. We speculate that except charge carriers but atomic displacement would be generated in polycrystalline CBD PbS film by the ionizing radiation, which partly destroys lattice structure resulting in increased defects[35–37]. PbS CQDs are of large specific surface area and quasi-amorphous, of which the surface exists many unsaturated bonds and vacancies. The irradiation energy of X-rays probably promotes ligand migration[38] and leads to self-healing of PbS CQDs[39], as shown in Supplementary Fig. S14. To test the idea, the defect change of PbS CQDs film under X-ray irradiation were characterized by variable temperature conductance measurement (Supplementary Fig. S15). The defect depth of the PbS CQDs film decreases from 0.122 eV to 0.101 eV after X-ray irradiation. Furthermore, photoluminescence (PL) of the PbS CQDs film is significantly increased by ~2 times, and the full width at half-maximum (FWHM) of PL is significantly narrow after X-ray irradiation for 6 h (Supplementary Fig. S16), demonstrating the passivation effect of irradiation. The deep understanding of this positive effect needs further investigation.

Furthermore, we investigated the flexibility of our PbS CQDs photodiode, as shown in Supplementary Fig. S17. The flexible device was fixed between two clamping kit to adjust the bending curvature by adjusting the distance between two kits (Supplementary Fig. S17a). The photoresponse of the flexible device with different bending angles

from 0 to 60° is demonstrated in Supplementary Fig. S17b under 970 nm LED illumination with the power density of 3 mW cm$^{-2}$ at −1 V bias. The dark current of the device barely changes even under a large bending angle of 60°. The photocurrent of the photodiode almost keeps the same at small bending angles (≤30°) and slightly decreases by 5% at a bending angle of 60°, possibly due to the ITO breaking. Then we measured the temporal response of the photodiode after different bending cycles at a fixed angle of 30° as shown in Supplementary Fig. S17c. The performance of the device is almost unchanged after 200 times bending. The SEM images of PbS CQDs film in Supplementary Fig. S18a−c exhibit that the surface morphology of PbS CQD film is compact and flat before bending. After bending at 30° for 200 times, there is nearly no obvious crack in the film (Supplementary Fig. S18d, e), suggesting favorable bending tolerance of the PbS CQDs film.

Finally, we prepared a flexible PbS CQDs photodiode array to exhibit imaging applications under X-ray and Vis-NIR illumination. Figure 4a demonstrates the schematic diagram of the imaging process in transmission mode, containing X-ray, visible and infrared sources, a capsule as object and flexible PbS CQDs photodiode array as the detector. For X-ray, visible and NIR imaging, the flexible PbS CQDs photodiode array with 20 × 20 pixels of 0.9 × 0.9 mm$^2$ pixel area was used to acquire the image by collecting the photo-signal of each pixel

and constructing image array. The multispectral imaging was achieved by simply changing the light source and using the single PbS CQDs photodiode array.

The physical photographs of the capsule open and sealed are demonstrated in Fig. 4b. The iron wire and wrapped soft plastic were sealed in the capsule. The X-ray image clearly shows the iron wire inside the capsule in Fig. 4c, due to that X-ray absorption of metal is much stronger than the organic coating. Figure 4d exhibits a blue-range image of the capsule under 450 nm illumination, which shows only an outline of the capsule but no patches on the capsule compared to the photograph in Fig. 4b. This is because the absorption of the capsule shell to 450 nm light is too high to observe the inside of the capsule. The NIR image in Fig. 4e shows more information under 970 nm illumination due to stronger NIR transmission of the capsule shell than visible. The main components of the capsule shell are gelatin and pigment with weak absorption in 800–1000 nm region[40]. Therefore, the iron wire and soft plastic filled with black ink are clearly emerged inside the capsule under 970 nm illumination compared with the blue-range image.

The process of pixel-by-pixel image fusion is exhibited in Fig. 4f. The photocurrent imaging matrices under different light sources are normalized to eliminate the interference of light intensity. A pixel-level image fusion is obtained by making a

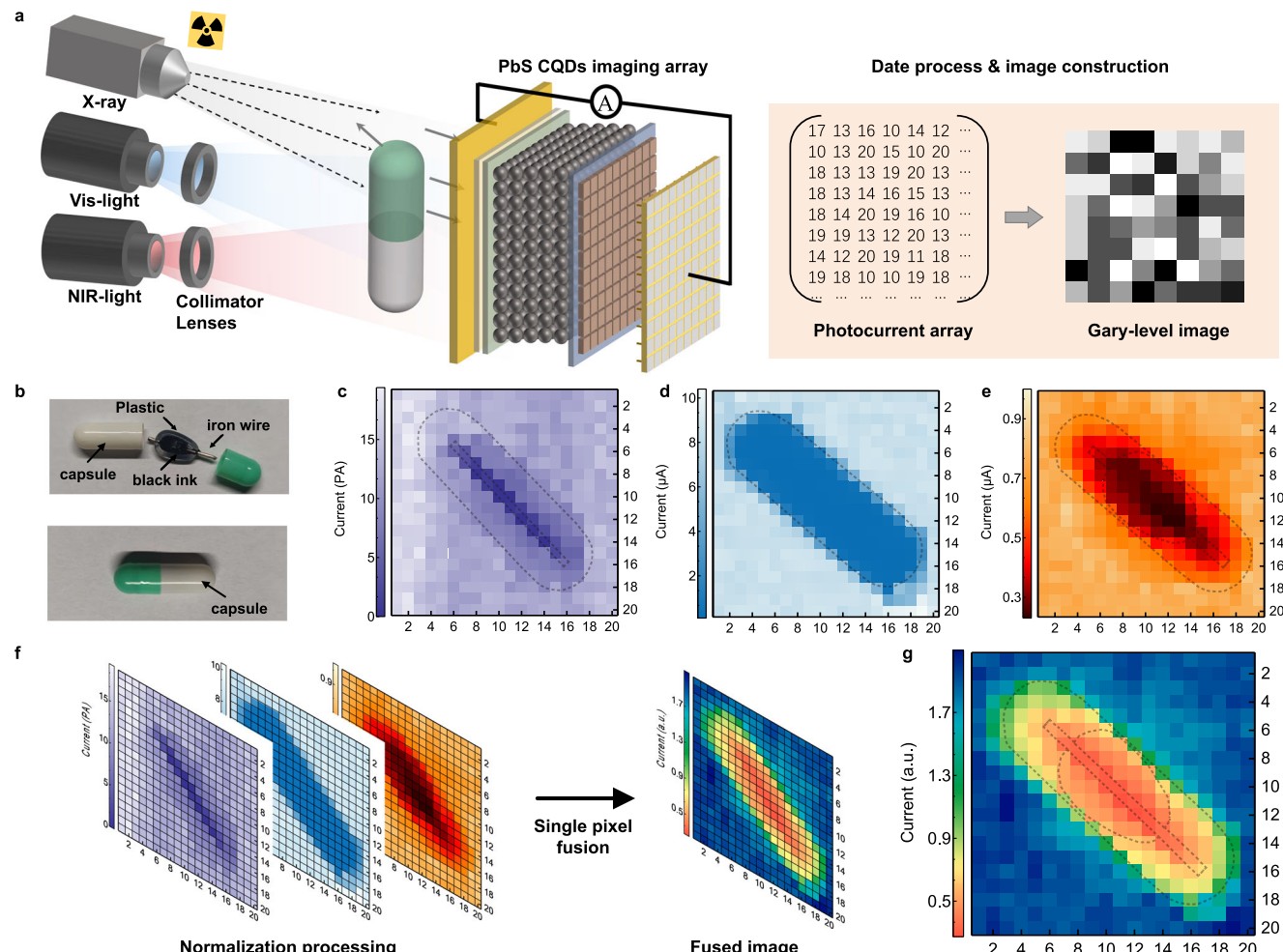

**Fig. 4 | Image fusion of flexible PbS CQDs photodiode array. a** Schematic illustration of X-ray, visible and infrared imaging process. The images were obtained by the same PbS CQDs photodiode array. **b** Photograph of a capsule inside and outside captured by a smartphone silicon imager. Soft plastic filled with black ink wrapping an iron wire is placed inside the capsule. Images of the capsule obtained by the photodiode array under X-ray (**c**), visible (450 nm, **d**) and NIR (970 nm, **e**) illumination. To guide the eye, black dotted lines show the outlines of capsule, soft plastic and iron wire. **f** Date process for pixel-level image fusion. **g** Fused image of X-ray, visible and NIR images.

weighted summation of the photocurrent array without pixel mismatch and complex imaging processing. The quality of the fused image is improved by optimizing the weight coefficients of X-ray, visible and NIR images in Supplementary Fig. S19. Since the NIR image contains more detailed information inside the capsule, the optimal fused image of X-ray, visible and NIR in Fig. 4g is obtained by increasing the weight value of NIR image. The detailed information of the capsule and its inclusions is comprehensively analyzed through image fusion from X-ray to NIR, which could be applied to clinical practice, such as blood vessels and skeleton. The single broadband PbS CQDs photodiode array exhibits the simple system for image fusion of X-ray, visible and NIR, providing great potential in a wide range of applications.

In summary, we prepared a flexible PbS CQDs photodiode array with ultra-broadband response from X-ray to NIR. The device demonstrated high responsivity and EQE ($0.38\,A\,W^{-1}$ and 76.6%), a low dark current density ($50.9\,nA\,cm^{-2}$), a large linear dynamic range (>85 dB), a high detectivity ($1.01\times10^{12}$ Jones) and a high $-3\,dB$ bandwidth (32 kHz) under Vis-NIR illumination. And the device showed $-17.8\,\mu C\,Gy_{air}^{-1}\,cm^{-2}$ sensitivity and $-2\times10^{5}\,\mu C\,Gy_{air}^{-1}\,cm^{-3}$ volume sensitivity under 30 keV X-ray irradiation at low bias (0.1–1.25 V). Furthermore, the device demonstrated favorable flexibility with almost no performance degradation after large-curvature (30°–60°) bending for 200 times. Finally, the single PbS CQDs photodiode array was used for pixel-level image fusion of X-ray, visible and NIR. The fused image can collect more comprehensive information from the different-band images without pixel mismatch and complex imaging processing. The simple method for pixel-level image fusion of X-ray, visible and NIR by our flexible PbS CQDs photodiode array demonstrated promising applications in flexible electronics.

## Methods

### Synthesis of PbS CQDs and ligand exchange

PbS CQDs were synthesized by the cation-exchange method as previously published in the article[29]. $PbCl_2$ was dissolved with oleylamine and octadecene (ODE) in a three-necked flask. The solution was heated and stirred to transparent and pre-prepared CdS CQDs were injected into the flash for Pb-Cd cation exchange. The resulting PbS CQDs were washed via precipitation with acetone and redispersion in octane two times. Solution-phase ligand exchange was processed in air with less than 50% humidity. 0.1 M $PbI_2$ and 0.04 M $PbBr_2$ were dissolved in DMF as the ligand solution[26]. A 10 mL PbS CQDs (20 mg/mL in octane) was added into 10 mL ligand solution. The mixed solution was oscillated vigorously for 1–2 min until the CQDs were completely transferred to the DMF phase. Then, the CQDs in DMF were washed for three times with octane. After ligand exchange, the CQDs were precipitated by adding ethyl acetate and separated by centrifugation. After drying in a vacuum, the PbS CQDs were re-dispersed in a mixed solvent (volume ratio of butylamine/amylamine/hexylamine as 10/3/2) with the concentration of 450 mg/mL.

### Device fabrication

The PbS CQDs photodiode arrays were fabricated on the polystyrene naphthalate (PEN) flexible substrate pre-deposited ITO with a square resistance of 7.2–9.5 Ohm/sq. A 40 nm hole transport layer of $NiO_x$ film was deposited by magnetron sputtering at 100 °C with the radiofrequency power of 200 W. Two layers of 1,2-ethanedithiol (EDT) treated PbS CQDs were deposited layer-by-layer with the thickness of 30–40 nm to reduce interface defects[38]. An active layer of $PbX_2$ ($X = I$, Br) capped PbS CQDs was fabricated by spin-coating using the CQDs ink after ligand exchange at 2500 rpm for 60 s and then annealed at 85 °C for 10 min. The thickness of the $PbX_2$-capped PbS CQD film was about 900 nm. And a 15 nm $C_{60}$ interlayer was deposited by thermal evaporation. Then an electron transport layer of 120 nm ZnO was

deposited by radio-frequency (RF) magnetron sputtering with the power of 200 W at room temperature. Finally, the patterned top electrodes (Al or ITO) were made with a shadow mask by thermal evaporation or magnetron sputtering.

### Material characterization

The morphology of PbS CQDs film was measured by scanning electron microscope (SEM, FEI NovaNanoSEM450) and atomic force microscope (AFM, SPM-9700 Shimadzu Co.,Japan). Transmission electron microscopy (TEM) images were obtained using FEI Tecnai G2 20 microscope with a $LaB_6$ filament operated at 200 KV. The absorption spectra were measured by Shimadzu UV-3600 plus spectrophotometer. The crystal structure of PbS CQDs film was identified by X-ray diffraction (XRD, Philips, X pert pro MRD, Cu Kα radiation) with a step of 0.013°. The Fourier transform infrared (FTIR) spectra of PbS CQDs film were investigated using VERTEX 70 ATR-FTIR spectroscope (Bruker Co., Germany). The X-ray photoelectron spectroscopy (XPS) measurements were performed on AXIS-ULTRA DLD-600W Ultra spectrometer (Kratos Co. Japan).

### Device measurement under UV-Vis-NIR illumination

Current density–voltage ($J$–$V$) and current-time ($I$–$t$) curves were measured using a semiconductor device parameter analyzer (Agilent B1500). Near-infrared (NIR) illumination was generated from a light-emitting diode with a wavelength of 970 nm. External quantum efficiency (EQE) and responsivity ($R$) were measured by the semiconductor analyzer (Agilent B1500A) and IPCE Measurement System (ENLITECH QE-R) light source. Temporal response and $-3\,dB$ bandwidth were measured by a high-speed oscilloscope (Agilent DSOS054A). A 780 nm LED was driven by a function generator (Agilent 33611) as a light source. Noise current was directly measured with a lock-in amplifier (SR850, Stanford Research Systems, Inc.). Shot noise, generation-recombination (G-R) noise and 1/$f$ noise were calculated by using the equation as described in the literature[41,42].

The linear dynamic range (LDR) was calculated by

$$LDR = 20\,\log(P_{max}/P_{min}) \qquad (1)$$

where $P_{max}$ and $P_{min}$ were the maximum and minimum optical power out of the photocurrent-power linear range. And the responsivity $R$ was defined by

$$R = I_{Ph}/P \qquad (2)$$

where $I_{ph}$ was the net photocurrent and $P$ was the optical power. The relationship between $R$ and EQE was given by

$$EQE = R/(hc/q\lambda) \qquad (3)$$

where $\lambda$ was the light wavelength, $h$ was the Planck constant, and c was the velocity of light. Specific detectivity ($D^*$) was calculated by

$$D^* = \frac{R\sqrt{A\Delta f}}{I_n} \qquad (4)$$

where $R$ was the responsivity, $A$ was the active area, $\Delta f$ was the electrical bandwidth (0.5 Hz) and $I_n$ was the measured noise current. And noise equivalent power (NEP) was given by

$$NEP = \frac{\sqrt{A\Delta f}}{D^*} \qquad (5)$$

where $A$ was the active area, $\Delta f$ was the electrical bandwidth (0.5 Hz) and $D^*$ was the specific detectivity.

## Device measurement under X-ray irradiation

The mass attenuation coefficients for PbS, Si, α-Se and CsPbBr$_3$ were calculated according to ESTAR/XCOM database[31]. The X-ray tube (HAMAMATSU L9421-02) with tungsten anode was used as the source. The X-ray focal spot size was 5 μm, which was operated with constant 50 keV acceleration voltage and 1–160 μA current to adjust the emitted X-ray dose rate. The X-ray dose rate was calibrated with the Fluke Si diode (RaySafe X2 R/F) and Radcal ion chamber (model:10 × 6–180) dosimeter. The $J–V$ and $I–t$ curves were measured by using the Keithley 4200-SCS semiconductor characterization system. The sensitivity of the detector is defined as the collected charge per unit area per unit exposure of radiation for a pixel assumed to receive the radiation[33]. The X-ray sensitivity was calculated by

$$S = I_{Ph}/D \tag{6}$$

where $I_{ph}$ was the net photocurrent and $D$ was the dose rate of X-ray.

## UV-Vis-NIR imaging

The UV-Vis-NIR images were obtained by the flexible PbS CQDs photodiode array with $20 \times 20$ pixels of $0.9 \times 0.9$ mm$^2$ area. The imaging system consisted of the LED light sources, a collimating lens and a semiconductor analyzer (Agilent B1500A). For imaging, the 450 nm and 780 nm LED were used as the light source with the power density of ~1 mW cm$^{-2}$ and placed about 2 cm away from the sample. And the photodiode array was closed to the object. The photocurrent of each pixel was collected and processed to construct images.

## X-ray imaging

The X-ray imaging system consisted of X-ray tube (HAMAMATSU L9421-02) and semiconductor analyzer (Keithley 4200-SCS) and control software. The X-ray tube was placed about 1.2 m away from the CQDs photodiode array and the object was located in the middle. Immobilizing the CQDs photodiode array, X-ray source and the sample, the photocurrent of each pixel was collected. Then, the record data array was used to construct images.

## Imaging fusion

The photocurrent matrices under different light sources were 8-bit normalized in a range of 0–1. The imaging matrices were obtained by weighted summation of the normalized photocurrent matrices pixel by pixel. The quality of the fused image could be improved by optimizing the weight factors of X-ray, visible and NIR photocurrent matrices. For images in this paper, the optimal weight factors of X-ray, visible and NIR photocurrent matrices were, respectively, 0.25, 0.125, and 0.625.

## Data availability

All other data that support the plots within this paper and other findings of this study are available from the corresponding authors upon request. Source data are provided with this paper.

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

## Acknowledgements

This work was financially supported by the National Key Research and Development Program of China (2021YFA0715502, J.T.), the National Natural Science Foundation of China (No. 62204091, X.L.), Key R & D program of Hubei Province (2021BAA014, L.G.), International science and technology cooperation project of Hubei Province (2021EHB010, L.G.), the Exploration Project of Natural Science Foundation of Zhejiang Province (No. LY23F040005, L.G.), the fund for Innovative Research Groups of the Natural Science Foundation of Hubei Province (No. 2020CFA034, J.T.), Scientific Research Project of Wenzhou (G20210013, L.G.), the Fund from Science, Technology and Innovation Commission of Shenzhen Municipality (No. GJHZ20210705142540010, J.Z.), and the China Postdoctoral Science Foundation (No. 2021M691118, S.L., 2022M711237, J.L.), the Innovation Project of Optics Valley Laboratory (No. OVL2021BG009, X.L., OVL2023ZD002, L.G.). The authors thank the Analytical and Testing Center of Huazhong University Science and Technology for the characterization.

## Author contributions

L.G. and J.T. supervised the whole project and revised the paper. J.L. specialized in the investigation, device fabrication, characterization, analysis, and writing. P.L. and T.S. helped to fabricate the device. M.K. and K.X. conducted the CQDs synthesis. Y.L. assisted in the imaging test. L.C. assisted in the device aging treatment. L.Z. helped to fabricate functional layers. X.L. and S.L. helped with LDR and variable temperature conductance measurements. H.L. help to optimize fusion image. G.N., J.Z., X.L., and P.F. helped to discuss the whole work.

## Competing interests
The authors declare no competing interests.
