## [Peer Review File · Nature Communications]

REVIEWER COMMENTS

Reviewer #1 (Remarks to the Author):

This work proposed an ideal solution to achieve pixel-level multispectral image fusion by flexible and broadband colloidal quantum dots photodiode array. It is a comprehensive work starting from the image fusion design to device performance measurements and strengthened by the pixel-level image fusion from X-ray to infrared. The experiments are delicately designed, and the conclusions are well supported. The performance of the photodiode array for Vis-NIR and X-ray are comparable to those of the commercial InGaAs (NIR) and α -Se (X-ray) detectors, suggesting great potential in flexible electronics. Overall, I found that the results are very solid, and the concepts are new, it should be published in this journal after addressing some minor issues.

1. As far as we know, the CQDs have more complicated surface states, which may have less radiation hardness than their bulk counterpart. Why do PbS CQDs have much better X-ray robustness compared to their bulk counterpart?
2. In the abstract, the authors claim that the X-ray sensitivity is $2 \times 10^4 \mu\text{C Gy}^{-1} \text{cm}^{-3}$, but the sensitivity is $17.8 \mu\text{C Gy}^{-1} \text{cm}^{-2}$ in the introduction, please clarify.
3. In Figure 2i, why the detector with different bias has the same 3dB frequency.
4. In the article, the thickness of the detector is only 900 nm, why not increase the film thickness to enhance X-ray absorption.?
5. In Figure 2, the PbS CQD-EDT layer and C60 layer were labeled in energy band diagram (2c), but not in 2a and 2b.
6. Details of the image fusion process need to be added like what weight factor was used.

Reviewer #2 (Remarks to the Author):

The authors provide well-quantified measurements that show that a colloidal solid composed of PbS quantum dots can exhibit x-ray, visible, and NIR performance metrics that are comparable or superior to other direct-detection technologies.

1) Abstract: "Image fusion extracts and combines information from multispectral images into a fused image, which is informative and beneficial for human or machine perception. However, currently multiple photodetectors with different response bands are used, which require complicated algorithm and system to solve the pixel and position mismatch problem."

The text could use a good grammar edit throughout. For instance, the second sentence of the above should be written "Currently, (you don't need the however) multiple photodetectors with different response regimes are used, which requires complicated algorithms and systems to solve the ..." (pluralize "algorithm" and "system"). Even the first sentence is redundant "Image fusion into a fused image....".... Instead, I would suggest "Combining information from multispectral images into a fused image is informative and beneficial for human or machine perception. (or some such)" Anyway, I won't English edit the rest of the paper but suggest you have someone do that (especially, pluralizing the various nouns throughout the paper).

2) Fig. 1: "Multi-scale tansform" typo (should be "Multi-scale transform")

3) Intro, pg 2: "Fusing X-ray, visible and infrared images as one single image could effectively and comprehensively construct the whole medical atlas as realized by the traditional approach (Fig. 1a) using three individual photodetectors for X-ray, visible, infrared and then applying vision algorithm."

Utilizing the same pixels for all wavelength bands can make the fused-image formation more computationally straightforward, but you should also comment on any performance costs associated with using the same readout plane. For instance: larger pixels for x-rays needed compared to visible in order to increase detection efficiency because of the far lower photon fluence of the source; secondary electron escape from x-ray-induced photoelectrons if the pixel size is too small; potential loss of NIR and visible image fidelity because of needs of x-ray imager. Is the cost in performance of using a single readout structure sufficiently small that the computational image processing gains compensate?

4) Intro, pg. 4: "Van der Waals interaction between adjacent dots allows slipping of CQDs without broken bonds and new defects under bending state (Fig. 1e), which supports desirable flexibility of CQDs devices." (Just for you information) even if the CQDs are chemically bonded (via oriented attachment for instance), the radius of curvature between neighboring QDs is sufficiently small (for small particles) that large scale macroscopic bending is possible.

5) Intro, pg. 4: "The pixel photodiode obtained impressive performance with a low dark current density (50.9 nA cm⁻² at -1 V)". That can be a large leakage current, depending on the PbS size. You might want to mention the QD diameter and size-dependent band-gap here so that the leakage current number can be understood as "impressive" in context, rather than simply in a supplementary figure.

6) Results and Discussion, pg. 4: “The as-prepared flexible 100×100 PbS CQDs photodiode array in the inset of Fig. 2a shows 20×20 mm² active area with 0.9×0.9 mm² pixel area and 0.1 mm pixel pitch patterned by a shadow mask.” Why did you choose this pixel size (very large for optical camera image)?

7) Fig. 2S caption: “Fig. S2 | Transmittance of flexible substrate and transport layers. a, Transmittance of ZnO film with thickness of 120 nm. b, Transmittance of NiOx film with thickness of 40 nm.” How were the oxide transport layer thicknesses optimized or chosen?

8) Results: pg. 5: “...adequate X-ray absorption. The active layer of PbS CQDs was fabricated by spin-coating with a thickness of ~900 nm.” Please define your definition of “adequate x-ray absorption”. How does the x-ray response (in whatever metric) vary for a greater or reduced number of layer-by-layer depositions?

9) Result, pg. 5: “The energy band alignment of PbS CQDs photodiode in Fig. 2c promotes efficient extraction of photo-generated electrons and holes and reduces recombination at electrodes.” Did you study the performance effect of altering the QD size in order to modify the alignment on the valence band? From Fig. 2c, it looks like a slightly smaller QD may improve the alignment.

10) Results, pg. 6: “The optimum EQE and R are 76.6% and 0.38 A W⁻¹ respectively at the wavelength of 620 nm.... The EQE and R are as high as 43.4% and 0.34 A W⁻¹ at 970 nm.”. What limits your EQE? Have you measured or calculated the expected photon detection probability across the wavelength bands? If the 900 nm thickness limits the EQE, what is the cost of making it thicker? This will help with x-ray response as well.

11) Results, pg. 6: “The rise time and fall time are respectively 4.8 and 5.1 μs, defined as time interval between 90% and 10% maximum photocurrent. And the rise time and fall time at -1 V bias are similarly 4.9 and 5.2 μs (supplementary Fig. S6) due to depleted junction.” Can you project the charge mobilities from these numbers or other measures? What is the charge transport mechanism?

12) Table S1: Supplementary Table S1 summarizes the performance of previously reported flexible photodetectors. Hmmm, it looks like the dark current of 50 nA/cm² is impressive compared to other flexible solids.

13) Result, pg. 8: “PbS with higher absorption coefficient allows thinner film to achieve adequate

X-ray absorption.”. You should mention though that the effective density of your QD film is less than the bulk and the polycrystalline film presumably.

14) Results, pg. 9: “The irradiation energy of Xrays probably promotes ligand migration 38 and leads to self-healing of PbS CQDs39 as shown in supplementary Fig. S11. To test the idea, the defect change of PbS CQDs film under X-ray irradiation were characterized by variable temperature conductance measurement (supplementary Fig. S12). The defect depth of the PbS CQDs film decreases from 0.122 eV to 0.101 eV after X-ray irradiation..... The deep understanding of this positive effect needs further investigation.” Yes on the last question, but these are nice measurements. However, why did you limit the stability study to short times (minutes or hours)... How is the stability over many days or weeks?

15) Results, pg. 9: “...and slightly decreases by 5% at bending angle of 60° possibly due to the ITO breaking.” Did you ensure that the exposed surface area is the same?

16) Fig. S16 captions: “The weight values of X-ay image, visible image and NIR image are both 0.3.” Change “both” to “all”. (“both” implies two images instead of three).

Reviewer #1 (Remarks to the Author):

This work proposed an ideal solution to achieve pixel-level multispectral image fusion by flexible and broadband colloidal quantum dots photodiode array. It is a comprehensive work starting from the image fusion design to device performance measurements and strengthened by the pixel-level image fusion from X-ray to infrared. The experiments are delicately designed, and the conclusions are well supported. The performance of the photodiode array for Vis-NIR and X-ray are comparable to those of the commercial InGaAs (NIR) and α -Se (X-ray) detectors, suggesting great potential in flexible electronics. Overall, I found that the results are very solid, and the concepts are new, it should be published in this journal after addressing some minor issues.

1. As far as we know, the CQDs have more complicated surface states, which may have less radiation hardness than their bulk counterpart. Why do PbS CQDs have much better X-ray robustness compared to their bulk counterpart?

Response: Thanks the reviewer for this helpful comment. As shown in supplementary Fig. S9b, continuous X-ray irradiation leads to an obviously improved dark current of bulk PbS photoconductor. X-ray photons acting on semiconductors would generate vacancies, interstitials and dislocations of atoms. The corresponding defects deteriorate the photoelectric properties of semiconductors. [doi.10.1149/2162-8777/abfc23]. For polycrystalline bulk PbS, atomic displacement would be generated by X-ray irradiation, which partly destroys lattice structure resulting in increased defects.

Surprisingly, continuous X-ray irradiation leads to decreased dark current and improved photocurrent of PbS CQDs photoconductor as shown in Fig. 3f, and the photoresponse tends to be stabilized after X-ray irradiation for 200 min. Different from polycrystalline bulk PbS, PbS CQDs are of large specific surface area and quasi-amorphous, of which the surface exists many unsaturated bonds and vacancies. There are many halide ligands around PbS CQDs in film, which are prone to ion migration under high-energy stimulations [doi.org/10.1002/adma.201702905]. We speculate that the irradiation energies of X-ray photons promote halide ion migration to passivate

unsaturated bonds and sulfur vacancies. The decreased defects of PbS CQDs after X-ray irradiation are also verified by temperature-dependent defect analysis (supplementary Fig. S15) and photoluminescence measurement (supplementary Fig. S16). Overall, this result is very interesting, and worthy further investigation.

2. In the abstract, the authors claim that the X-ray sensitivity is $2 \times 10^4 \mu\text{C Gy}^{-1} \text{cm}^{-3}$, but the sensitivity is $17.8 \mu\text{C Gy}^{-1} \text{cm}^{-2}$ in the introduction, please clarify.

Response: We are thankful for the reviewer's comment. We revised the description of $2 \times 10^4 \mu\text{C Gy}^{-1} \text{cm}^{-3}$ as volume sensitivity and $17.8 \mu\text{C Gy}^{-1} \text{cm}^{-2}$ as area sensitivity.

(Line 28-29, Page 1)

We revised the manuscript accordingly “The CQDs photodiode array showed a specific detectivity exceeding 10^{12} Jones in visible and NIR range and a favorable volume sensitivity of approximately $2 \times 10^4 \mu\text{C Gy}^{-1} \text{cm}^{-3}$ for X-ray irradiation.”

(Line 21, Page 4)

We revised the manuscript accordingly “It could be operated at a very low voltage (0.1-1.25 V) with an area sensitivity of $17.8 \mu\text{C Gy}^{-1} \text{cm}^{-2}$...”.

3. In Figure 2i, why the detector with different bias has the same -3dB frequency.

Response: Thanks the reviewer for raising this concern. The response time of PbS CQDs photodetectors is limited by various factors including drift time, diffusion time, and RC (resistor-capacitor) time. The photocurrent of the device saturates at a low reverse bias (-0.5 V), revealing the PbS CQDs layer would be completely depleted under bias of -0.5 to -2 V as shown in Fig. 2d. Therefore, the response time of our PbS CQDs devices is determined by the drift time and RC time. According to the previous report, as the active area of CQDs device decreases, the response rate significantly increases. Therefore, the RC time primarily limits the response rate of the CQDs photodetector. [doi.org/10.1016/j.matt.2020.12.017] The -3dB frequency of our CQDs devices is mainly limited by geometrical capacitance rather than bias voltage.

4. In the article, the thickness of the detector is only 900 nm, why not increase the film thickness to enhance X-ray absorption?

Response: Thanks the reviewer for raising this concern. 900 nm is a balanced thickness for our detector considering its NIR and X-ray detection performance. We made PbS CQDs photodiodes with different thickness of CQDs layer and added their photoresponses as supplementary Fig. S5. Thicker CQDs layer enhances X-ray and NIR absorption. The high penetration depth of X-ray enables photogenerated carriers within or near the depleted region, which facilitates effective extraction of photogenerated carriers. The photoresponse to X-ray is enhanced by increasing the thickness of CQDs layer. However, for NIR illumination, the photogenerated carriers are mainly at the surface of CQDs layer far from the depletion region, resulting in low extraction efficiency and hence lower performance. Considering the contradictory requirement, 900 nm is the optimal thickness for our device.

(Supporting Information)

Fig. S5| Photoresponses of PbS CQDs photodiodes with different thickness of PbS CQDs layer. a, Transient responses at -0.1 V bias under 5.1 mGy_{air} s⁻¹ dose rates X-ray. b, Transient responses at -0.1 V bias illuminated by 970 nm LED with a power

density of 0.45 mW cm^{-2} . Photogenerated carrier transmission in CQDs photodiodes under (c) X-ray and (d) Vis-NIR irradiation³.

5. In Figure 2, the PbS CQD-EDT layer and C60 layer were labeled in energy band diagram (2c), but not in 2a and 2b.

Response: We are thankful for the reviewer's reminding. We added clear labels in Figure 2a and 2b.

(Line 1, Page 20)

We revised Figure 2 accordingly.

6. Details of the image fusion process need to be added like what weight factor was used.

Response: We are thankful for the reviewer's reminding. We added the detailed information of the image fusion process in Materials and Methods.

(Line 13-19, Page 15)

We revised the manuscript accordingly.

Imaging fusion

The photocurrent matrices under different light sources were 8-bit normalized in a range of 0-1. The imaging matrices were obtained by weighted summation of the normalized photocurrent matrices pixel by pixel. The quality of fused image could be improved by optimizing the weight factors of X-ray, visible and NIR photocurrent matrices. For images in this paper, the optimal weight factors of X-ray, visible and NIR photocurrent matrices were respectively 0.25, 0.125 and 0.625.

Reviewer #2 (Remarks to the Author):

The authors provide well-quantified measurements that show that a colloidal solid composed of PbS quantum dots can exhibit x-ray, visible, and NIR performance metrics that are comparable or superior to other direct-detection technologies.

1. Abstract: “Image fusion extracts and combines information from multispectral images into a fused image, which is informative and beneficial for human or machine perception. However, currently multiple photodetectors with different response bands are used, which require complicated algorithm and system to solve the pixel and position mismatch problem.” The text could use a good grammar edit throughout. For instance, the second sentence of the above should be written “Currently, (you don’t need the however) multiple photodetectors with different response regimes are used, which requires complicated algorithms and systems to solve the ...” (pluralize “algorithm” and “system”). Even the first sentence is redundant “Image fusion into a fused image....”.... Instead, I would suggest “Combining information from multispectral images into a fused image is informative and beneficial for human or machine perception. (or some such)” Anyway, I won’t English edit the rest of the paper but suggest you have someone do that (especially, pluralizing the various nouns throughout the paper).

Response: We are thankful for the reviewer’s suggestions. We polished the manuscript and pluralize the various nouns throughout the paper as below.

(Line 18, Page 1-Line 2, Page 2)

We revised the manuscript accordingly.

Combining information from multispectral images into a fused image is informative and beneficial for human or machine perception. Currently, multiple photodetectors with different response bands are used, which require complicated algorithms and systems to solve the pixel and position mismatch problem. An ideal solution would be pixel-level multispectral image fusion (PLMSIF), which involves multispectral image using the same photodetector and circumventing the mismatch problem. Here we presented the potential of PLMSIF utilizing colloidal quantum dots (CQDs) photodiode array, with a broadband response range from X-ray to near infrared (NIR) and excellent tolerance for bending and X-ray irradiation. The CQDs photodiode array showed a

specific detectivity exceeding 10^{12} Jones in visible and NIR range and a favorable volume sensitivity of approximately $2 \times 10^4 \mu\text{C Gy}^{-1} \text{cm}^{-3}$ for X-ray irradiation. To showcase the advantages of PLMIF, we imaged a capsule enfolding an iron wire and soft plastic, successfully revealing internal information through an X-ray to NIR fused image.

(Line 5, Page 2-Line 27, Page 3)

We revised the manuscript accordingly.

Multi-spectral image fusion is a technique that extracts the most pertinent information from different-wavelength source images into a unified image, with the goal of providing richer and more valuable information for subsequent applications, such as machine vision¹, autonomous vehicles², medical diagnosis³ and other artificial intelligences⁴. Existing approaches for multi-spectral image fusion typically rely on vision algorithms, including multi-scale transformation⁵, deep learning⁶ and *etc.*, at the sacrifice of resolution mismatch, overloaded computing resources and complicated systems⁷. With the advancement of photodetectors that have broader response range, pixel-level image fusion can be a more practical approach, where multi-spectral images are captured using just one photodetector. This approach simplifies imaging processes and systems, with the additional benefits of conserving computational resources and reducing energy consumption. For example, traditional InGaAs photodetectors have been modified to broaden their response range from 0.9–1.7 μm to 0.4–1.7 μm for visible-infrared pixel-level image fusion⁸, yielding more informative images in the inclement weather.

Pixel-level multi-spectral image fusion (PLMSIF) of X-ray, visible and infrared is highly desired in various areas such as medical imaging⁹, security monitoring¹⁰ and nondestructive testing¹¹. As for application in medical imaging, the X-ray image emphasizes the inorganic skeleton texture, while the visible image supports the assessment of appearance, and the infrared image provides a detailed description of organic tissue structure. Combining X-ray, visible and infrared images into one single image can effectively and comprehensively construct the complete medical atlas, as

realized by the traditional approach (**Fig. 1a**) using three individual photodetectors for X-ray, visible, infrared and then applying a vision algorithm. This system requires complex vision algorithms and extensive computing resources to compensate for the differences in pixel position and resolution between the three types of photodetectors, impeding the development of artificial intelligence in medical imaging. As another increasingly active demand for comfortable and real-time medical imaging, wearable and flexible photodetectors also need to be taken into consideration and developed to fit irregular biology surface and improve comfort level. However, as far as we are concerned, there is no report on one single flexible photodetector capable of capturing X-ray, visible and infrared images to achieve image fusion (**Fig. 1b**). This new approach is very appropriate for flexible lensless imaging, such as biomedical measurement and medical diagnosis¹².

Various materials such as halide perovskites^{12, 13}, organic semiconductors¹⁴, two-dimensional materials^{15, 16} and colloidal quantum dots (CQDs)^{17, 18} have emerged, enabling flexible and wide detection range beyond traditional silicon and InGaAs photodetectors. Halide perovskites are ultra-sensitive and have a low detection limit for X-ray and visible detection due to their high absorption coefficient and high $\mu\tau$ product, but they show poor performance for infrared detection owing to their large bandgap^{19, 20}. Organic semiconductors have achieved ultra-low dark current, large linear dynamic range and excellent flexibility but with limited response range and poor X-ray irradiation resistance²¹. Two-dimensional materials such as graphene exhibit fast photoresponse and ultra-broadband response from visible to terahertz, but they are too thin to efficiently absorb X-ray and have limited capacities for imaging array²². PbS CQDs are widely recognized for their excellent visible and infrared photodetection capabilities, which are attributed to their tunable bandgap, high absorption coefficient and low-temperature solution processing²³⁻²⁵. Actually, these materials contain heavy element Pb which is a strong absorber for X-ray because the absorption coefficient of X-ray is proportional to the fourth power of atomic number (Pb, 82). Furthermore, as shown in our manuscript, PbS CQDs exhibit much better X-ray robustness compared

to their bulk counterpart. Hence, PbS CQDs are at least one of the best choices for the pixel-level X-ray to infrared image fusion.

2. Fig. 1: “Multi-scale tansform” typo (should be “Multi-scale transform”)

Response: We are thankful for the reviewer’s reminding. We corrected the error in Fig.1.

(Line 1, Page 19)

3. Intro, pg 2: “Fusing X-ray, visible and infrared images as one single image could effectively and comprehensively construct the whole medical atlas as realized by the traditional approach (Fig. 1a) using three individual photodetectors for X-ray, visible, infrared and then applying vision algorithm.” Utilizing the same pixels for all

wavelength bands can make the fused-image formation more computationally straightforward, but you should also comment on any performance costs associated with using the same readout plane. For instance: larger pixels for x-rays needed compared to visible in order to increase detection efficiency because of the far lower photon fluence of the source; secondary electron escape from x-ray-induced photoelectrons if the pixel size is too small; potential loss of NIR and visible image fidelity because of needs of x-ray imager. Is the cost in performance of using a single readout structure sufficiently small that the computational image processing gains compensate?

Response: Thanks the reviewer for raising this concern. In this work, we propose a new approach to simplify the complex computational processes during multispectral image fusion. Considering far lower photon fluence and much weaker convergence of the X-ray source, the commercial X-ray imaging system has large pixel size and no lens. Similar to the commercial X-ray system, our imaging system also has large pixel size, which is beneficial to sensitive photoresponses to X-ray, visible and NIR light. If used for optical camera with lens, our imaging system needs expensive large-aperture lens. Hence, our approach to fuse X-ray, visible and NIR images by one single photodetector is appropriate for flexible lens-free imaging, such as biomedical measurement and medical diagnosis [doi.org/10.1038/s41928-019-0354-7].

(Line 6-8, Page 3)

We revised the manuscript accordingly “This new approach could be useful for flexible lens-free imaging, such as biomedical measurement and medical diagnosis¹².”

4. Intro, pg. 4: “Van der Waals interaction between adjacent dots allows slipping of CQDs without broken bonds and new defects under bending state (Fig. 1e), which supports desirable flexibility of CQDs devices.” (Just for you information) even if the CQDs are chemically bonded (via oriented attachment for instance), the radius of curvature between neighboring QDs is sufficiently small (for small particles) that large scale macroscopic bending is possible.

Response: Thanks the reviewer for raising the discussion. We agree with your viewpoint. We calculated the strain of bended PbS CQDs film and added the detailed description in the article as below.

(Line 13, Page 4)

We revised the manuscript accordingly “Van der Waals interaction between adjacent dots allows slipping of CQDs without broken bonds and new defects under bending state (Fig. 1e), which supports desirable flexibility of CQDs devices (Supplementary Fig. S2)”

(Supporting Information)

Fig. S2| a, Schematic diagram of bended PbS CQDs photodiode. **b,** Strain as a function of bending curvature.

Analysis of strain within a bended device is shown in Fig. S2a. As the PbS CQDs device is one thousand times thinner than the PI substrate, the neutral plane with zero strain is situated on the surface of the PI substrate¹. The strain ϵ_z at different positions can be rewritten as below

$$\epsilon_z = \frac{z - z_{NA}}{r}$$

where Z is the location of the CQDs device, Z_{NA} is the location of neutral plane, r is the curvature radius of film. r can be calculated by the equation:

$$r = \frac{360^\circ}{4\theta} \times \frac{l}{2\pi}$$

where l is the length of the CQDs device, θ is the bending angle. Thus, we can obtain the relationship between θ and ε_z as shown in Fig. S2b. The maximum strain of PbS CQDs device is 0.15% at a bending angle of 90° . According to the previous report, the average inter-dot spacing is ~ 3.21 nm². The inter-dot spacing of PbS CQDs only changes 0.00045 nm. It is possible to achieve a high degree of curvature even when the PbS CQDs are chemically bonded.

5. Intro, pg. 4: “The pixel photodiode obtained impressive performance with a low dark current density (50.9 nA cm⁻² at -1 V)”. That can be a large leakage current, depending on the PbS size. You might want to mention the QD diameter and size-dependent band-gap here so that the leakage current number can be understood as “impressive” in context, rather than simply in a supplementary figure.

Response: We are thankful for the reviewer’s remind. We added the detailed description in the article as below.

(Line 6-7, Page 6)

We revised the manuscript accordingly “The device exhibits a low dark current density as 50.9 nA/cm² at -1 V bias and a high rectification ratio of around 1000 at ± 1 V bias, where the bandgap of our PbS CQDs is 1.18 eV.”

6. Results and Discussion, pg. 4: “The as-prepared flexible 100×100 PbS CQDs photodiode array in the inset of Fig. 2a shows 20×20 mm² active area with 0.9×0.9 mm² pixel area and 0.1 mm pixel pitch patterned by a shadow mask.” Why did you choose this pixel size (very large for optical camera image)?

Response: Thanks the reviewer for raising this concern. In this work, we present the design of a simple large-area imaging system to assess the feasibility of capturing multiple images using a single photodetector. The design of this imaging system mainly refers to the commercial X-ray thin-film-transistor (TFT) detector array. The commercial a-Se flat panel X-ray detectors (e.g. Hologic and ANRAD) typically have over 100 μ m pixel size [doi.org/10.3390/qubs5040029]. In order to achieve better X-

ray imaging, we designed a larger pixel size of 900 μm to increase X-ray absorption and hence improve the X-ray response. The pixel size can be reduced for higher-resolution lens-free imaging and further for the optical camera with lens. In addition, this lens-free imaging system with large pixel size is very suitable for biomedical measurements, venous imaging and medical diagnostic as photons are very limited under these scenarios. [doi.10.1038/ncomms6745]

7. Fig. 2S caption: “Fig. S2| Transmittance of flexible substrate and transport layers. a, Transmittance of ZnO film with thickness of 120 nm. b, Transmittance of NiO_x film with thickness of 40 nm.” How were the oxide transport layer thicknesses optimized or chosen?

Response: Thanks the reviewer for raising this concern. We characterized the transport properties of ZnO and NiO_x layers using Hall measurements as below. The halide capped PbS CQDs film is P-type doped and its carrier concentration is about $\sim 10^{16} \text{ cm}^{-3}$ according to the literature [doi.org/10.1038/s41467-019-13158-6]. The width of the depletion region (X_D) can be determined using the equation

$$X_D = \sqrt{\frac{2\epsilon_r\epsilon_0(N_A + N_D)(V_D - V)}{qN_A N_D}}$$

where N_A and N_D are the carrier concentration of PbS CQDs and ZnO, V_D is the contact potential difference, V is the bias Voltage, ϵ_r is the relative dielectric constant, ϵ_0 is the absolute dielectric constant, q is the electron charge. [doi.10.1002/adfm.201804502]

The X_D of ZnO/PbS CQDs heterojunction are approximately 370 nm at zero bias, 600 nm at -1 V, 770 nm at -2 V, and 900 nm at -3 V. The depletion width in the n-type ZnO (x_n) and p-type PbS CQDs (x_p) layers can be calculated using the following formula:

$$x_n = \frac{N_A X_D}{N_D + N_A}, x_p = \frac{N_D X_D}{N_D + N_A}$$

The calculated maximum depletion width in ZnO layer is approximately ~ 90 nm. We experimentally determined the optimal thickness of the ZnO layer to be 120 nm as shown in Fig. R1a. The primary function of NiO_x is to act as an electron blocking layer, which can reduce carrier recombination. But its deep valence band maximum forms

hole transport barrier that hinders the extraction of holes as shown in Fig. 2c. The optimal thickness of the NiO_x layer is about 40 nm through the J - V tests (Fig. R1b).

Table R1. Parameters of the ZnO and NiO_x layers in optimal PbS CQDs device.

	Thickness (nm)	Mobility (cm ² V ⁻¹ s ⁻¹)	Carrier density (cm ⁻³)
NiO _x	40	1.33 ± 0.2	3.6(± 2.1) × 10 ¹⁶
ZnO	120	0.11 ± 0.3	1.0(± 3.1) × 10 ¹⁷

Fig. R1. Current-voltage (J - V) curves under dark and 970 nm LED illumination with different thickness of ZnO (a) and NiO_x (b).

8. Results: pg. 5: “...adequate X-ray absorption. The active layer of PbS CQDs was fabricated by spin-coating with a thickness of ~900 nm.” Please define your definition of “adequate x-ray absorption”. How does the x-ray response (in whatever metric) vary for a greater or reduced number of layer-by-layer depositions?

Response: Thanks the reviewer for raising this concern. The absorption efficiency of PbS to 50 keV X-ray photon versus thickness is shown in Fig. S11a. As the film's thickness increases, the X-ray absorption efficiency steadily increases until it reaches 90% at a thickness of ~400 μm. We made PbS CQDs photodiodes with different thickness of CQDs layer and added their photoresponses as supplementary Fig. S5. Thicker CQDs layer enhances X-ray and NIR absorption. The high penetration depth of X-ray enables photogenerated carriers within or near the depleted region, which facilitates effective extraction of photogenerated carriers. The photoresponse to X-ray

is enhanced by increasing the thickness of CQDs layer. However, the photogenerated carriers by NIR illumination are mainly at the surface of CQDs layer, which is outside the depletion region and hence suffers from with low extraction efficiency. The photoresponse to NIR is optimal when the CQDs thickness is 900 nm. When the CQDs thickness exceeds the optimal value (~ 900 nm), incomplete carrier extraction causes a severe drop in EQE to NIR.

(Line 13-15, Page 5)

We revised the manuscript accordingly “The active layer of PbS CQDs was fabricated by spin-coating with a thickness of ~ 900 nm enabling $\sim 5\%$ X-ray absorption (supplementary Fig. S5 and S11).”

(Supporting Information)

Fig. S5| Photoresponses of PbS CQDs photodiodes with different thickness of PbS CQDs layer. a, Transient responses at -0.1 V bias under $5.1 \text{ mGy}_{\text{air}} \text{ s}^{-1}$ dose rates X-ray. **b,** Transient responses at -0.1 V bias illuminated by 970 nm LED with a power density of 0.45 mW cm^{-2} . Photogenerated carrier transmission in CQDs photodiodes under (c) X-ray and (d) Vis-NIR irradiation³.

9. Result, pg. 5: “The energy band alignment of PbS CQDs photodiode in Fig. 2c promotes efficient extraction of photo-generated electrons and holes and reduces recombination at electrodes.” Did you study the performance effect of altering the QD size in order to modify the alignment on the valence band? From Fig. 2c, it looks like a slightly smaller QD may improve the alignment.

Response: Thanks the reviewer for raising this concern. The energy band structure of PbS CQDs is demonstrated in Fig. R2a as a function of CQD diameter [doi.org/10.1021/nn201681s]. And the energy band structure of PbS CQDs is also affected by ligands [doi.org/10.1021/nn500897c]. As shown in Fig. 2c, there is a typical hole transport layer of ethanedithiol-treated CQDs with a larger bandgap (1.41 eV) than the halide-passivated CQDs active layer (1.32 eV), which enables efficient hole extraction and electron blocking. We used larger-size CQDs with smaller bandgap as active layer to fabricate CQDs photodiodes. The dark and photo $J-V$ curves in Fig. R2b show that as the CQDs size increases, the carrier extraction is still efficient due to the matched band energy alignment.

Fig. R2. a, Energy band structure of PbS CQDs as a function of CQD diameter [doi.org/10.1021/nn201681s]. b, Dark and photo $J-V$ curves of CQDs photodiodes with active layers of different-size CQDs.

10. Results, pg. 6: “The optimum EQE and R are 76.6% and 0.38 A W^{-1} respectively at the wavelength of 620 nm.... The EQE and R are as high as 43.4% and 0.34 A W^{-1} at

970 nm.”. What limits your EQE? Have you measured or calculated the expected photon detection probability across the wavelength bands? If the 900 nm thickness limits the EQE, what is the cost of making it thicker? This will help with x-ray response as well.

Response: Thanks the reviewer for raising this concern. The EQE of photodiodes is limited not just by absorption efficiency of light-absorbing layer, but also by extraction efficiency of photogenerated carriers. As shown in Fig. R3, the photogenerated carriers are extracted from the diffusion and drift regions [doi.org/10.21203/rs.3.rs-677155/v1]. The carrier diffusion length in PbS CQDs film is about 150-250 nm, because of low mobility (10^{-4} - 10^{-3} $\text{cm}^2 \text{V}^{-1} \text{s}^{-1}$) [doi.10.1038/NPHOTON.2015.280]. When the thickness of CQDs layer is exceeding 900 nm, the extraction efficiency of photogenerated carriers is reduced, further limiting the EQE of the CQDs photodiodes (Fig. S5). We are working on improving the mobility of our CQD film so that thicker film could be used for better X-ray and NIR detection performance.

Fig. R3. Photogenerated carrier transmission in PbS CQDs photodiode under NIR irradiation [DOI:10.1038/s41928-022-00779-x].

11. Results, pg. 6: “The rise time and fall time are respectively 4.8 and 5.1 μs , defined as time interval between 90% and 10% maximum photocurrent. And the rise time and fall time at -1 V bias are similarly 4.9 and 5.2 μs (supplementary Fig. S6) due to depleted junction.” Can you project the charge mobilities from these numbers or other measures? What is the charge transport mechanism?

Response: Thanks the reviewer for raising this concern. The response time of photodiodes is limited by various factors including drift time, diffusion time, and RC

(resistor-capacitor) time. According to the previous report, as the active area of CQDs photodiode decreases, the response rate significantly increases. Hence, the RC time primarily defines the response time of the CQDs photodetector when the active area is over 0.01 mm². [doi.org/10.1016/j.matt.2020.12.017] As the mobility has little relation with RC time, we couldn't project the charge mobility of CQDs from the rise and fall time.

The charge transport mechanism in CQDs film is that carriers transfer to adjacent CQDs by tunneling. [doi.org/10.1021/jz300048y] The tunneling probability depends on the spacing between adjacent CQDs and ligand type. As the spacing between adjacent CQDs decreases, the wavefunction of the electrons becomes more overlapping, resulting in higher tunneling probability and better mobility [doi.10.1021/acs.nanolett.6b04201]. The field-effect transistor (FET) test is commonly used for measuring the mobility of CQDs film [doi.10.1038/s41427-020-0215-x, doi.10.1038/s41467-018-06342-7]. The mobility of CQDs film can be calculated from the slope of the transfer curve of CQDs FET. We carried out FET measurements to characterize the carrier mobility as shown in Fig. R4a. The carrier mobility (μ) is calculated according to the equation

$$\mu = \frac{L}{C_i W V_{DS}} \cdot \frac{I_D}{V_G - V_{TH}}$$

where I_D is the drain current, L and W are the channel length (10 μm) and channel width (180 μm) respectively, V_G and V_{TH} are the gate voltage and threshold voltage, C_i is the capacitance per unit area of the dielectric layer. The mobility of PbS CQDs film is measured as $\sim 4.63 \times 10^{-3} \text{ cm}^2/\text{V}\cdot\text{s}$ (Fig. R4b).

Fig. R4. a, Schematic diagram of the CQDs FET. b, Transfer characteristics of PbS CQDs FET.

12. Table S1: Supplementary Table S1 summarizes the performance of previously reported flexible photodetectors. Hmmm, it looks like the dark current of 50 nA/cm² is impressive compared to other flexible solids.

Response: We are thankful for the reviewer’s comment. We supplemented the dark current density of the flexible photodetectors under different operating biases in the supporting information.

(Supporting Information)

Table S1| Performance list of flexible photodetectors.

material	Spectral range (nm)	R (A/W ⁻¹)/EQE (%)	Dark current density (nA/cm ²)	Detectivity (Jones)	Response time (s)	LDR (dB)	Ref.
Sb ₂ Se ₃	450–1050	0.42 83%	~900 (-0.1 V) ~2000 (-1 V) ^[C]	2.4×10 ¹¹ [M]	1.6×10 ⁻⁶	95	1
organic	350–1000	1200%	~50 (-1 V) ~100 (-5 V) ^[C]	2.0×10 ¹² [M]	/	158	4
MAPbI ₃	300–800	0.4 75%	~34 (-0.1 V)	1.1×10 ¹⁰ [M]	9.8×10 ⁻⁷	112	5
ZnO/PbS CQDs	350–1100	4.54	/	3.98×10 ¹² [C]	1.01	>60	6
PbS CQDs	350–1300	0.38 76.6%	12.6 (-0.1 V) 50.9 (-1 V) V	1.01×10 ¹² [M]	5×10 ⁻⁶	>85	This work

‘C’ means the dark current density calculated from the given dark current and device area in the article.

13. Result, pg. 8: “PbS with higher absorption coefficient allows thinner film to achieve adequate X-ray absorption.”. You should mention though that the effective density of your QD film is less than the bulk and the polycrystalline film presumably.

Response: We are thankful for the reviewer’s comment. We supplemented the description of the effective density of PbS CQDs film in the article as below. Based on the energy dispersive spectroscopy (EDS) results presented in Table S2, the photon cross section for X-ray absorption of PbS CQDs can be estimated from the mass percentages of Pb, S, I, and Br elements in the film. According to the dense stacking model [doi.org/10.1038/s41563-019-0582-2], the CQDs film contains 64% volume ratio of PbS CQDs and 36% volume ratio of PbI_2 and $PbBr_2$ [doi.org/10.1021/acs.nanolett.1c01892]. The mass absorption coefficient of the PbS CQD film was calculated based on this model as shown in Fig. 3a.

(Line 2-3, 7-8, Page 8)

We revised the manuscript accordingly “...which is higher than some typical semiconductors such as Si and α -Se on account of its large average atomic number. ... Bulk PbS and PbS CQDs with higher absorption coefficient than traditional Si and α -Se allow thinner film to achieve adequate X-ray absorption.”

(Line 1, Page 21)

Fig. 3. Performance of flexible PbS CQDs photodiode under X-ray irradiation. a, Mass absorption coefficient of bulk PbS, PbS CQDs, Si, α -Se and CsPbBr₃ as a function of photon energy. **b,** Current density-voltage curves of the PbS CQDs photodiode under dark and X-ray irradiation with 8.1 mGy_{air} s⁻¹ dose rates. **c,** Photocurrent and sensitivity to X-ray varying with the dose rate at -1 V bias. **d,** Transient response to X-ray under different dose rates at -1 V bias. **e,** Transient response at various biases under 8.1 mGy_{air} s⁻¹ dose rates. **f,** X-ray irradiation stability of PbS CQDs film.

(Supporting Information)

Table S2| Energy dispersive spectroscopy (EDS) results of the PbS CQDs film.

Element	Line type	Mass percent (%)	Atomic percent (%)
Pb	M	64.78	38.46
I	L	20.51	19.88
S	K	8.28	31.77
Br	L	6.42	9.89

14. Results, pg. 9: “The irradiation energy of X rays probably promotes ligand migration 38 and leads to self-healing of PbS CQDs³⁹ as shown in supplementary Fig. S11. To test the idea, the defect change of PbS CQDs film under X-ray irradiation were characterized by variable temperature conductance measurement (supplementary Fig. S12). The defect depth of the PbS CQDs film decreases from 0.122 eV to 0.101 eV after X-ray irradiation..... The deep understanding of this positive effect needs further investigation.” Yes on the last question, but these are nice measurements. However, why did you limit the stability study to short times (minutes or hours)... How is the stability over many days or weeks?

Response: We are thankful for the reviewer’s comment. We monitored the photoresponse of PbS CQDs film under X-ray irradiation (5.5 mGy_{air} s⁻¹) for longer times. We supplemented the stability of PbS CQDs film under X-ray irradiation in the article as below. The photoresponse of 7 PbS CQDs films remains stable under X-ray

irradiation for one week.

(Line 10-12, Page 9)

We revised the manuscript accordingly “The photoresponse of 7 PbS CQDs films remains stable after X-ray irradiation ($5.5 \text{ mGyair s}^{-1}$) for one week with a total dosage of 3326 Gyair dosage (supplementary Fig. S13).”

(Supporting Information)

We revised the supporting information file accordingly.

Fig. S13| Photoresponse of 7 PbS CQDs films under X-ray irradiation.

15. Results, pg. 9: “...and slightly decreases by 5% at bending angle of 60° possibly due to the ITO breaking.” Did you ensure that the exposed surface area is the same?

Response: Thanks the reviewer for raising this concern. We bent the CQDs photodiode at various angles and then released it to original flat state for the photoresponse tests. Hence, the exposed surface area is the same in the photoresponse tests. Through morphology characterization as shown in Fig. R5, we observed stripped cracks on the surface of the ITO film after 60° bending. We suspected that the slight degradation of device performance was due to ITO damage after 60° bending.

Fig. R5. SEM image of ITO film after 60° bending

16. Fig. S16 captions: “The weight values of X-ray image, visible image and NIR image are both 0.3.” Change “both” to “all”. (“both” implies two images instead of three).

Response: We are thankful for the reviewer’s reminding. We corrected the error in the captions.

(Supporting Information)

We revised the supporting information file accordingly “Fig. S19| Fused images with different weights coefficients. a, The weight values of X-ray image, visible image and NIR image are **all 0.3.**”

REVIEWER COMMENTS

Reviewer #3 (Remarks to the Author):

The manuscript by Liu et al. presented their research on flexible detectors based on PbS colloidal quantum dots. The significant achievements here are a single detector can be used for UV, VIS, NIR and especially X-ray (thanks to Pb content in the QDs); then demonstration for multispectral image fusion with a detector array is compelling. I acknowledge the hard work and nice results of the research but I can not justify it being published in Nature Communications due to the following reasons.

1. The detector structure is standard, and performance is not superior either; many demonstrations have been demonstrated already. Very quick search, we can find PbS QD photodetectors with a responsivity of 373 A/W and a detectivity of 10^{13} Jones (Nanotechnology 32 195502) much better than the current manuscript. The X-ray response is stated as “compete well with the reported X-ray direct detectors” but the reference is from 2003. How can it compare with new results such as Nat Commun 9, 2926 (2018)? The possible significance here might be the array structure and X-ray detection with a photodetector device. But an array is just an incremental engineering demonstration, and I have no doubt that previous PbS photodetector devices in literature respond to X-rays as well.
2. 900 nm thickness of PbS is stated to be determined by the diffusion and drift length of photogenerated carriers and adequate X-ray absorption. This statement is very standard, all researchers know such information but how to get 900nm is a mystery. Is it really optimized or simply a one-shot?
3. Basically, I did not learn much new knowledge from this manuscript rather than seeing a fancy demonstration, which is worth publishing but in a specialized journal.

Reviewer #1 (Remarks to the Author):

This work proposed an ideal solution to achieve pixel-level multispectral image fusion by flexible and broadband colloidal quantum dots photodiode array. It is a comprehensive work starting from the image fusion design to device performance measurements and strengthened by the pixel-level image fusion from X-ray to infrared. The experiments are delicately designed, and the conclusions are well supported. The performance of the photodiode array for Vis-NIR and X-ray are comparable to those of the commercial InGaAs (NIR) and α -Se (X-ray) detectors, suggesting great potential in flexible electronics. Overall, I found that the results are very solid, and the concepts are new, it should be published in this journal after addressing some minor issues.

1. As far as we know, the CQDs have more complicated surface states, which may have less radiation hardness than their bulk counterpart. Why do PbS CQDs have much better X-ray robustness compared to their bulk counterpart?

Response: Thanks the reviewer for this helpful comment. As shown in supplementary Fig. S9b, continuous X-ray irradiation leads to an obviously improved dark current of bulk PbS photoconductor. X-ray photons acting on semiconductors would generate vacancies, interstitials and dislocations of atoms. The corresponding defects deteriorate the photoelectric properties of semiconductors. [doi.10.1149/2162-8777/abfc23]. For polycrystalline bulk PbS, atomic displacement would be generated by X-ray irradiation, which partly destroys lattice structure resulting in increased defects.

Surprisingly, continuous X-ray irradiation leads to decreased dark current and improved photocurrent of PbS CQDs photoconductor as shown in Fig. 3f, and the photoresponse tends to be stabilized after X-ray irradiation for 200 min. Different from polycrystalline bulk PbS, PbS CQDs are of large specific surface area and quasi-amorphous, of which the surface exists many unsaturated bonds and vacancies. There are many halide ligands around PbS CQDs in film, which are prone to ion migration under high-energy stimulations [doi.org/10.1002/adma.201702905]. We speculate that the irradiation energies of X-ray photons promote halide ion migration to passivate

unsaturated bonds and sulfur vacancies. The decreased defects of PbS CQDs after X-ray irradiation are also verified by temperature-dependent defect analysis (supplementary Fig. S15) and photoluminescence measurement (supplementary Fig. S16). Overall, this result is very interesting, and worthy further investigation.

2. In the abstract, the authors claim that the X-ray sensitivity is $2 \times 10^5 \mu\text{C Gy}^{-1} \text{cm}^{-3}$, but the sensitivity is $17.8 \mu\text{C Gy}^{-1} \text{cm}^{-2}$ in the introduction, please clarify.

Response: We are thankful for the reviewer's comment. We revised the description of $2 \times 10^5 \mu\text{C Gy}^{-1} \text{cm}^{-3}$ as volume sensitivity and $17.8 \mu\text{C Gy}^{-1} \text{cm}^{-2}$ as area sensitivity.

(Line 28-29, Page 1)

We revised the manuscript accordingly “The CQDs photodiode array showed a specific detectivity exceeding 10^{12} Jones in visible and NIR range and a favorable volume sensitivity of approximately $2 \times 10^5 \mu\text{C Gy}^{-1} \text{cm}^{-3}$ for X-ray irradiation.”

(Line 21, Page 4)

We revised the manuscript accordingly “It could be operated at a very low voltage (0.1-1.25 V) with an area sensitivity of $17.8 \mu\text{C Gy}^{-1} \text{cm}^{-2}$...”.

3. In Figure 2i, why the detector with different bias has the same -3dB frequency.

Response: Thanks the reviewer for raising this concern. The response time of PbS CQDs photodetectors is limited by various factors including drift time, diffusion time, and RC (resistor-capacitor) time. The photocurrent of the device saturates at a low reverse bias (-0.5 V), revealing the PbS CQDs layer would be completely depleted under bias of -0.5 to -2 V as shown in Fig. 2d. Therefore, the response time of our PbS CQDs devices is determined by the drift time and RC time. According to the previous report, as the active area of CQDs device decreases, the response rate significantly increases. Therefore, the RC time primarily limits the response rate of the CQDs photodetector. [doi.org/10.1016/j.matt.2020.12.017] The -3dB frequency of our CQDs devices is mainly limited by geometrical capacitance rather than bias voltage.

4. In the article, the thickness of the detector is only 900 nm, why not increase the film thickness to enhance X-ray absorption?

Response: Thanks the reviewer for raising this concern. 900 nm is a balanced thickness for our detector considering its NIR and X-ray detection performance. We made PbS CQDs photodiodes with different thickness of CQDs layer and added their photoresponses as supplementary Fig. S5. Thicker CQDs layer enhances X-ray and NIR absorption. The high penetration depth of X-ray enables photogenerated carriers within or near the depleted region, which facilitates effective extraction of photogenerated carriers. The photoresponse to X-ray is enhanced by increasing the thickness of CQDs layer. However, for NIR illumination, the photogenerated carriers are mainly at the surface of CQDs layer far from the depletion region, resulting in low extraction efficiency and hence lower performance. Considering the contradictory requirement, 900 nm is the optimal thickness for our device.

(Supporting Information)

Fig. S5| Photoresponse of PbS CQDs photodiodes with different thickness of PbS CQDs layer. a, Transient responses at -0.1 V bias under $5.1 \text{ mGy}_{\text{air}} \text{ s}^{-1}$ dose rates X-ray. b, Transient responses at -0.1 V bias illuminated by 970 nm LED with a power

density of 0.45 mW cm^{-2} . Photogenerated carrier transmission in CQDs photodiodes under (c) X-ray and (d) Vis-NIR irradiation³.

5. In Figure 2, the PbS CQD-EDT layer and C60 layer were labeled in energy band diagram (2c), but not in 2a and 2b.

Response: We are thankful for the reviewer's reminding. We added clear labels in Figure 2a and 2b.

(Line 1, Page 20)

We revised Figure 2 accordingly.

6. Details of the image fusion process need to be added like what weight factor was used.

Response: We are thankful for the reviewer's reminding. We added the detailed information of the image fusion process in Materials and Methods.

(Line 13-19, Page 15)

We revised the manuscript accordingly.

Imaging fusion

The photocurrent matrices under different light sources were 8-bit normalized in a range of 0-1. The imaging matrices were obtained by weighted summation of the normalized photocurrent matrices pixel by pixel. The quality of fused image could be improved by optimizing the weight factors of X-ray, visible and NIR photocurrent matrices. For images in this paper, the optimal weight factors of X-ray, visible and NIR photocurrent matrices were respectively 0.25, 0.125 and 0.625.

Reviewer #2 (Remarks to the Author):

The authors provide well-quantified measurements that show that a colloidal solid composed of PbS quantum dots can exhibit x-ray, visible, and NIR performance metrics that are comparable or superior to other direct-detection technologies.

1. Abstract: “Image fusion extracts and combines information from multispectral images into a fused image, which is informative and beneficial for human or machine perception. However, currently multiple photodetectors with different response bands are used, which require complicated algorithm and system to solve the pixel and position mismatch problem.” The text could use a good grammar edit throughout. For instance, the second sentence of the above should be written “Currently, (you don’t need the however) multiple photodetectors with different response regimes are used, which requires complicated algorithms and systems to solve the ...” (pluralize “algorithm” and “system”). Even the first sentence is redundant “Image fusion into a fused image....”.... Instead, I would suggest “Combining information from multispectral images into a fused image is informative and beneficial for human or machine perception. (or some such)” Anyway, I won’t English edit the rest of the paper but suggest you have someone do that (especially, pluralizing the various nouns throughout the paper).

Response: We are thankful for the reviewer’s suggestions. We polished the manuscript and pluralize the various nouns throughout the paper as below.

(Line 18, Page 1-Line 2, Page 2)

We revised the manuscript accordingly.

Combining information from multispectral images into a fused image is informative and beneficial for human or machine perception. Currently, multiple photodetectors with different response bands are used, which require complicated algorithms and systems to solve the pixel and position mismatch problem. An ideal solution would be pixel-level multispectral image fusion (PLMSIF), which involves multispectral image using the same photodetector and circumventing the mismatch problem. Here we presented the potential of PLMSIF utilizing colloidal quantum dots (CQDs) photodiode array, with a broadband response range from X-ray to near infrared (NIR) and excellent tolerance for bending and X-ray irradiation. The CQDs photodiode array showed a

specific detectivity exceeding 10^{12} Jones in visible and NIR range and a favorable volume sensitivity of approximately $2 \times 10^4 \mu\text{C Gy}^{-1} \text{cm}^{-3}$ for X-ray irradiation. To showcase the advantages of PLMIF, we imaged a capsule enfolding an iron wire and soft plastic, successfully revealing internal information through an X-ray to NIR fused image.

(Line 5, Page 2-Line 27, Page 3)

We revised the manuscript accordingly.

Multi-spectral image fusion is a technique that extracts the most pertinent information from different-wavelength source images into a unified image, with the goal of providing richer and more valuable information for subsequent applications, such as machine vision¹, autonomous vehicles², medical diagnosis³ and other artificial intelligences⁴. Existing approaches for multi-spectral image fusion typically rely on vision algorithms, including multi-scale transformation⁵, deep learning⁶ and *etc.*, at the sacrifice of resolution mismatch, overloaded computing resources and complicated systems⁷. With the advancement of photodetectors that have broader response range, pixel-level image fusion can be a more practical approach, where multi-spectral images are captured using just one photodetector. This approach simplifies imaging processes and systems, with the additional benefits of conserving computational resources and reducing energy consumption. For example, traditional InGaAs photodetectors have been modified to broaden their response range from 0.9–1.7 μm to 0.4–1.7 μm for visible-infrared pixel-level image fusion⁸, yielding more informative images in the inclement weather.

Pixel-level multi-spectral image fusion (PLMSIF) of X-ray, visible and infrared is highly desired in various areas such as medical imaging⁹, security monitoring¹⁰ and nondestructive testing¹¹. As for application in medical imaging, the X-ray image emphasizes the inorganic skeleton texture, while the visible image supports the assessment of appearance, and the infrared image provides a detailed description of organic tissue structure. Combining X-ray, visible and infrared images into one single image can effectively and comprehensively construct the complete medical atlas, as

realized by the traditional approach (**Fig. 1a**) using three individual photodetectors for X-ray, visible, infrared and then applying a vision algorithm. This system requires complex vision algorithms and extensive computing resources to compensate for the differences in pixel position and resolution between the three types of photodetectors, impeding the development of artificial intelligence in medical imaging. As another increasingly active demand for comfortable and real-time medical imaging, wearable and flexible photodetectors also need to be taken into consideration and developed to fit irregular biology surface and improve comfort level. However, as far as we are concerned, there is no report on one single flexible photodetector capable of capturing X-ray, visible and infrared images to achieve image fusion (**Fig. 1b**). This new approach is very appropriate for flexible lensless imaging, such as biomedical measurement and medical diagnosis¹².

Various materials such as halide perovskites^{12, 13}, organic semiconductors¹⁴, two-dimensional materials^{15, 16} and colloidal quantum dots (CQDs)^{17, 18} have emerged, enabling flexible and wide detection range beyond traditional silicon and InGaAs photodetectors. Halide perovskites are ultra-sensitive and have a low detection limit for X-ray and visible detection due to their high absorption coefficient and high $\mu\tau$ product, but they show poor performance for infrared detection owing to their large bandgap^{19, 20}. Organic semiconductors have achieved ultra-low dark current, large linear dynamic range and excellent flexibility but with limited response range and poor X-ray irradiation resistance²¹. Two-dimensional materials such as graphene exhibit fast photoresponse and ultra-broadband response from visible to terahertz, but they are too thin to efficiently absorb X-ray and have limited capacities for imaging array²². PbS CQDs are widely recognized for their excellent visible and infrared photodetection capabilities, which are attributed to their tunable bandgap, high absorption coefficient and low-temperature solution processing²³⁻²⁵. Actually, these materials contain heavy element Pb which is a strong absorber for X-ray because the absorption coefficient of X-ray is proportional to the fourth power of atomic number (Pb, 82). Furthermore, as shown in our manuscript, PbS CQDs exhibit much better X-ray robustness compared

to their bulk counterpart. Hence, PbS CQDs are at least one of the best choices for the pixel-level X-ray to infrared image fusion.

2. Fig. 1: “Multi-scale tansform” typo (should be “Multi-scale transform”)

Response: We are thankful for the reviewer’s reminding. We corrected the error in Fig.1.

(Line 1, Page 19)

3. Intro, pg 2: “Fusing X-ray, visible and infrared images as one single image could effectively and comprehensively construct the whole medical atlas as realized by the traditional approach (Fig. 1a) using three individual photodetectors for X-ray, visible, infrared and then applying vision algorithm.” Utilizing the same pixels for all

wavelength bands can make the fused-image formation more computationally straightforward, but you should also comment on any performance costs associated with using the same readout plane. For instance: larger pixels for x-rays needed compared to visible in order to increase detection efficiency because of the far lower photon fluence of the source; secondary electron escape from x-ray-induced photoelectrons if the pixel size is too small; potential loss of NIR and visible image fidelity because of needs of x-ray imager. Is the cost in performance of using a single readout structure sufficiently small that the computational image processing gains compensate?

Response: Thanks the reviewer for raising this concern. In this work, we propose a new approach to simplify the complex computational processes during multispectral image fusion. Considering far lower photon fluence and much weaker convergence of the X-ray source, the commercial X-ray imaging system has large pixel size and no lens. Similar to the commercial X-ray system, our imaging system also has large pixel size, which is beneficial to sensitive photoresponses to X-ray, visible and NIR light. If used for optical camera with lens, our imaging system needs expensive large-aperture lens. Hence, our approach to fuse X-ray, visible and NIR images by one single photodetector is appropriate for flexible lens-free imaging, such as biomedical measurement and medical diagnosis [doi.org/10.1038/s41928-019-0354-7].

(Line 6-8, Page 3)

We revised the manuscript accordingly “This new approach could be useful for flexible lens-free imaging, such as biomedical measurement and medical diagnosis¹².”

4. Intro, pg. 4: “Van der Waals interaction between adjacent dots allows slipping of CQDs without broken bonds and new defects under bending state (Fig. 1e), which supports desirable flexibility of CQDs devices.” (Just for you information) even if the CQDs are chemically bonded (via oriented attachment for instance), the radius of curvature between neighboring QDs is sufficiently small (for small particles) that large scale macroscopic bending is possible.

Response: Thanks the reviewer for raising the discussion. We agree with your viewpoint. We calculated the strain of bended PbS CQDs film and added the detailed description in the article as below.

(Line 13, Page 4)

We revised the manuscript accordingly “Van der Waals interaction between adjacent dots allows slipping of CQDs without broken bonds and new defects under bending state (Fig. 1e), which supports desirable flexibility of CQDs devices (Supplementary Fig. S2)”

(Supporting Information)

Fig. S2| a, Schematic diagram of bended PbS CQDs photodiode. **b,** Strain as a function of bending curvature.

Analysis of strain within a bended device is shown in Fig. S2a. As the PbS CQDs device is one thousand times thinner than the PI substrate, the neutral plane with zero strain is situated on the surface of the PI substrate¹. The strain ϵ_z at different positions can be rewritten as below

$$\epsilon_z = \frac{z - z_{NA}}{r}$$

where Z is the location of the CQDs device, Z_{NA} is the location of neutral plane, r is the curvature radius of film. r can be calculated by the equation:

$$r = \frac{360^\circ}{4\theta} \times \frac{l}{2\pi}$$

where l is the length of the CQDs device, θ is the bending angle. Thus, we can obtain the relationship between θ and ε_z as shown in Fig. S2b. The maximum strain of PbS CQDs device is 0.15% at a bending angle of 90° . According to the previous report, the average inter-dot spacing is $\sim 3.21 \text{ nm}^2$. The inter-dot spacing of PbS CQDs only changes 0.00045 nm. It is possible to achieve a high degree of curvature even when the PbS CQDs are chemically bonded.

5. Intro, pg. 4: “The pixel photodiode obtained impressive performance with a low dark current density (50.9 nA cm^{-2} at -1 V)”. That can be a large leakage current, depending on the PbS size. You might want to mention the QD diameter and size-dependent band-gap here so that the leakage current number can be understood as “impressive” in context, rather than simply in a supplementary figure.

Response: We are thankful for the reviewer’s remind. We added the detailed description in the article as below.

(Line 6-7, Page 6)

We revised the manuscript accordingly “The device exhibits a low dark current density as 50.9 nA/cm^2 at -1 V bias and a high rectification ratio of around 1000 at $\pm 1 \text{ V}$ bias, where the bandgap of our PbS CQDs is 1.18 eV.”

6. Results and Discussion, pg. 4: “The as-prepared flexible 100×100 PbS CQDs photodiode array in the inset of Fig. 2a shows $20 \times 20 \text{ mm}^2$ active area with $0.9 \times 0.9 \text{ mm}^2$ pixel area and 0.1 mm pixel pitch patterned by a shadow mask.” Why did you choose this pixel size (very large for optical camera image)?

Response: Thanks the reviewer for raising this concern. In this work, we present the design of a simple large-area imaging system to assess the feasibility of capturing multiple images using a single photodetector. The design of this imaging system mainly refers to the commercial X-ray thin-film-transistor (TFT) detector array. The commercial a-Se flat panel X-ray detectors (e.g. Hologic and ANRAD) typically have over $100 \mu\text{m}$ pixel size [doi.org/10.3390/qubs5040029]. In order to achieve better X-

ray imaging, we designed a larger pixel size of 900 μm to increase X-ray absorption and hence improve the X-ray response. The pixel size can be reduced for higher-resolution lens-free imaging and further for the optical camera with lens. In addition, this lens-free imaging system with large pixel size is very suitable for biomedical measurements, venous imaging and medical diagnostic as photons are very limited under these scenarios. [doi.10.1038/ncomms6745]

7. Fig. 2S caption: “Fig. S2| Transmittance of flexible substrate and transport layers. a, Transmittance of ZnO film with thickness of 120 nm. b, Transmittance of NiOx film with thickness of 40 nm.” How were the oxide transport layer thicknesses optimized or chosen?

Response: Thanks the reviewer for raising this concern. We characterized the transport properties of ZnO and NiO_x layers using Hall measurements as below. The halide capped PbS CQDs film is P-type doped and its carrier concentration is about $\sim 10^{16} \text{ cm}^{-3}$ according to the literature [doi.org/10.1038/s41467-019-13158-6]. The width of the depletion region (X_D) can be determined using the equation

$$X_D = \sqrt{\frac{2\varepsilon_r\varepsilon_0(N_A + N_D)(V_D - V)}{qN_A N_D}}$$

where N_A and N_D are the carrier concentration of PbS CQDs and ZnO, V_D is the contact potential difference, V is the bias Voltage, ε_r is the relative dielectric constant, ε_0 is the absolute dielectric constant, q is the electron charge. [doi.10.1002/adfm.201804502]

The X_D of ZnO/PbS CQDs heterojunction are approximately 370 nm at zero bias, 600 nm at -1 V, 770 nm at -2 V, and 900 nm at -3 V. The depletion width in the n-type ZnO (x_n) and p-type PbS CQDs (x_p) layers can be calculated using the following formula:

$$x_n = \frac{N_A X_D}{N_D + N_A}, x_p = \frac{N_D X_D}{N_D + N_A}$$

The calculated maximum depletion width in ZnO layer is approximately ~ 90 nm. We experimentally determined the optimal thickness of the ZnO layer to be 120 nm as shown in Fig. R1a. The primary function of NiO_x is to act as an electron blocking layer, which can reduce carrier recombination. But its deep valence band maximum forms

hole transport barrier that hinders the extraction of holes as shown in Fig. 2c. The optimal thickness of the NiO_x layer is about 40 nm through the J - V tests (Fig. R1b).

Table R1. Parameters of the ZnO and NiO_x layers in optimal PbS CQDs device.

	Thickness (nm)	Mobility (cm ² V ⁻¹ s ⁻¹)	Carrier density (cm ⁻³)
NiO _x	40	1.33 ± 0.2	3.6(± 2.1) × 10 ¹⁶
ZnO	120	0.11 ± 0.3	1.0(± 3.1) × 10 ¹⁷

Fig. R1. Current-voltage (J - V) curves under dark and 970 nm LED illumination with different thickness of ZnO (a) and NiO_x (b).

8. Results: pg. 5: “...adequate X-ray absorption. The active layer of PbS CQDs was fabricated by spin-coating with a thickness of ~900 nm.” Please define your definition of “adequate x-ray absorption”. How does the x-ray response (in whatever metric) vary for a greater or reduced number of layer-by-layer depositions?

Response: Thanks the reviewer for raising this concern. The absorption efficiency of PbS to 50 keV X-ray photon versus thickness is shown in Fig. S11a. As the film's thickness increases, the X-ray absorption efficiency steadily increases until it reaches 90% at a thickness of ~400 μm. We made PbS CQDs photodiodes with different thickness of CQDs layer and added their photoresponses as supplementary Fig. S5. Thicker CQDs layer enhances X-ray and NIR absorption. The high penetration depth of X-ray enables photogenerated carriers within or near the depleted region, which facilitates effective extraction of photogenerated carriers. The photoresponse to X-ray

is enhanced by increasing the thickness of CQDs layer. However, the photogenerated carriers by NIR illumination are mainly at the surface of CQDs layer, which is outside the depletion region and hence suffers from with low extraction efficiency. The photoresponse to NIR is optimal when the CQDs thickness is 900 nm. When the CQDs thickness exceeds the optimal value (~ 900 nm), incomplete carrier extraction causes a severe drop in EQE to NIR.

(Line 13-15, Page 5)

We revised the manuscript accordingly “The active layer of PbS CQDs was fabricated by spin-coating with a thickness of ~ 900 nm enabling $\sim 5\%$ X-ray absorption (supplementary Fig. S5 and S11).”

(Supporting Information)

Fig. S5| Photoresponse of PbS CQDs photodiodes with different thickness of PbS CQDs layer. a, Transient responses at -0.1 V bias under $5.1 \text{ mGy}_{\text{air}} \text{ s}^{-1}$ dose rates X-ray. **b,** Transient responses at -0.1 V bias illuminated by 970 nm LED with a power density of 0.45 mW cm^{-2} . Photogenerated carrier transmission in CQDs photodiodes under (c) X-ray and (d) Vis-NIR irradiation³.

9. Result, pg. 5: “The energy band alignment of PbS CQDs photodiode in Fig. 2c promotes efficient extraction of photo-generated electrons and holes and reduces recombination at electrodes.” Did you study the performance effect of altering the QD size in order to modify the alignment on the valence band? From Fig. 2c, it looks like a slightly smaller QD may improve the alignment.

Response: Thanks the reviewer for raising this concern. The energy band structure of PbS CQDs is demonstrated in Fig. R2a as a function of CQD diameter [doi.org/10.1021/nn201681s]. And the energy band structure of PbS CQDs is also affected by ligands [doi.org/10.1021/nn500897c]. As shown in Fig. 2c, there is a typical hole transport layer of ethanedithiol-treated CQDs with a larger bandgap (1.41 eV) than the halide-passivated CQDs active layer (1.32 eV), which enables efficient hole extraction and electron blocking. We used larger-size CQDs with smaller bandgap as active layer to fabricate CQDs photodiodes. The dark and photo $J-V$ curves in Fig. R2b show that as the CQDs size increases, the carrier extraction is still efficient due to the matched band energy alignment.

Fig. R2. a, Energy band structure of PbS CQDs as a function of CQD diameter [doi.org/10.1021/nn201681s]. b, Dark and photo $J-V$ curves of CQDs photodiodes with active layers of different-size CQDs.

10. Results, pg. 6: “The optimum EQE and R are 76.6% and 0.38 A W^{-1} respectively at the wavelength of 620 nm.... The EQE and R are as high as 43.4% and 0.34 A W^{-1} at

970 nm.”. What limits your EQE? Have you measured or calculated the expected photon detection probability across the wavelength bands? If the 900 nm thickness limits the EQE, what is the cost of making it thicker? This will help with x-ray response as well.

Response: Thanks the reviewer for raising this concern. The EQE of photodiodes is limited not just by absorption efficiency of light-absorbing layer, but also by extraction efficiency of photogenerated carriers. As shown in Fig. R3, the photogenerated carriers are extracted from the diffusion and drift regions [doi.org/10.21203/rs.3.rs-677155/v1]. The carrier diffusion length in PbS CQDs film is about 150-250 nm, because of low mobility (10^{-4} - 10^{-3} cm² V⁻¹ s⁻¹) [doi.10.1038/NPHOTON.2015.280]. When the thickness of CQDs layer is exceeding 900 nm, the extraction efficiency of photogenerated carriers is reduced, further limiting the EQE of the CQDs photodiodes (Fig. S5). We are working on improving the mobility of our CQD film so that thicker film could be used for better X-ray and NIR detection performance.

Fig. R3. Photogenerated carrier transmission in PbS CQDs photodiode under NIR irradiation [DOI:10.1038/s41928-022-00779-x].

11. Results, pg. 6: “The rise time and fall time are respectively 4.8 and 5.1 μ s, defined as time interval between 90% and 10% maximum photocurrent. And the rise time and fall time at -1 V bias are similarly 4.9 and 5.2 μ s (supplementary Fig. S6) due to depleted junction.” Can you project the charge mobilities from these numbers or other measures? What is the charge transport mechanism?

Response: Thanks the reviewer for raising this concern. The response time of photodiodes is limited by various factors including drift time, diffusion time, and RC

(resistor-capacitor) time. According to the previous report, as the active area of CQDs photodiode decreases, the response rate significantly increases. Hence, the RC time primarily defines the response time of the CQDs photodetector when the active area is over 0.01 mm². [doi.org/10.1016/j.matt.2020.12.017] As the mobility has little relation with RC time, we couldn't project the charge mobility of CQDs from the rise and fall time.

The charge transport mechanism in CQDs film is that carriers transfer to adjacent CQDs by tunneling. [doi.org/10.1021/jz300048y] The tunneling probability depends on the spacing between adjacent CQDs and ligand type. As the spacing between adjacent CQDs decreases, the wavefunction of the electrons becomes more overlapping, resulting in higher tunneling probability and better mobility [doi.10.1021/acs.nanolett.6b04201]. The field-effect transistor (FET) test is commonly used for measuring the mobility of CQDs film [doi.10.1038/s41427-020-0215-x, doi.10.1038/s41467-018-06342-7]. The mobility of CQDs film can be calculated from the slope of the transfer curve of CQDs FET. We carried out FET measurements to characterize the carrier mobility as shown in Fig. R4a. The carrier mobility (μ) is calculated according to the equation

$$\mu = \frac{L}{C_i W V_{DS}} \cdot \frac{I_D}{V_G - V_{TH}}$$

where I_D is the drain current, L and W are the channel length (10 μm) and channel width (180 μm) respectively, V_G and V_{TH} are the gate voltage and threshold voltage, C_i is the capacitance per unit area of the dielectric layer. The mobility of PbS CQDs film is measured as $\sim 4.63 \times 10^{-3} \text{ cm}^2/\text{V}\cdot\text{s}$ (Fig. R4b).

Fig. R4. a, Schematic diagram of the CQDs FET. b, Transfer characteristics of PbS CQDs FET.

12. Table S1: Supplementary Table S1 summarizes the performance of previously reported flexible photodetectors. Hmmm, it looks like the dark current of 50 nA/cm² is impressive compared to other flexible solids.

Response: We are thankful for the reviewer’s comment. We supplemented the dark current density of the flexible photodetectors under different operating biases in the supporting information.

(Supporting Information)

Table S1| Performance list of flexible photodetectors.

material	Spectral range (nm)	R (A/W ⁻¹)/ EQE (%)	Dark current density (nA/cm ²)	Detectivity (Jones)	Response time (s)	LDR (dB)	Ref.
Sb ₂ Se ₃	450–1050	0.42 83%	~900 (-0.1 V) ~2000 (-1 V) ^[C]	2.4×10 ¹¹ [M]	1.6×10 ⁻⁶	95	1
organic	350–1000	1200%	~50 (-1 V) ~100 (-5 V) ^[C]	2.0×10 ¹² [M]	/	158	4
MAPbI ₃	300–800	0.4 75%	~34 (-0.1 V)	1.1×10 ¹⁰ [M]	9.8×10 ⁻⁷	112	5
ZnO/PbS CQDs	350–1100	4.54	/	3.98×10 ¹² [C]	1.01	>60	6
PbS CQDs	350–1300	0.38 76.6%	12.6 (-0.1 V) 50.9 (-1 V)	1.01×10 ¹² [M]	5×10 ⁻⁶	>85	This work

‘C’ means the dark current density calculated from the given dark current and device area in the article.

13. Result, pg. 8: “PbS with higher absorption coefficient allows thinner film to achieve adequate X-ray absorption.”. You should mention though that the effective density of your QD film is less than the bulk and the polycrystalline film presumably.

Response: We are thankful for the reviewer’s comment. We supplemented the description of the effective density of PbS CQDs film in the article as below. Based on the energy dispersive spectroscopy (EDS) results presented in Table S2, the photon cross section for X-ray absorption of PbS CQDs can be estimated from the mass percentages of Pb, S, I, and Br elements in the film. According to the dense stacking model [doi.org/10.1038/s41563-019-0582-2], the CQDs film contains 64% volume ratio of PbS CQDs and 36% volume ratio of PbI_2 and $PbBr_2$ [doi.org/10.1021/acs.nanolett.1c01892]. The mass absorption coefficient of the PbS CQD film was calculated based on this model as shown in Fig. 3a.

(Line 2-3, 7-8, Page 8)

We revised the manuscript accordingly “...which is higher than some typical semiconductors such as Si and α -Se on account of its large average atomic number. ... Bulk PbS and PbS CQDs (supplementary Table S2) with higher absorption coefficient than traditional Si and α -Se allow thinner film to achieve adequate X-ray absorption.”

(Line 1, Page 21)

Fig. 3. Performance of flexible PbS CQDs photodiode under X-ray irradiation. a, Mass absorption coefficient of bulk PbS, PbS CQDs, Si, α -Se and CsPbBr₃ as a function of photon energy. **b,** Current density-voltage curves of the PbS CQDs photodiode under dark and X-ray irradiation with 8.1 mGy_{air} s⁻¹ dose rates. **c,** Photocurrent and sensitivity to X-ray varying with the dose rate at -1 V bias. **d,** Transient response to X-ray under different dose rates at -1 V bias. **e,** Transient response at various biases under 8.1 mGy_{air} s⁻¹ dose rates. **f,** X-ray irradiation stability of PbS CQDs film.

(Supporting Information)

Table S2| Energy dispersive spectroscopy (EDS) results of the PbS CQDs film.

Element	Line type	Mass percent (%)	Atomic percent (%)
Pb	M	64.78	38.46
I	L	20.51	19.88
S	K	8.28	31.77
Br	L	6.42	9.89

14. Results, pg. 9: “The irradiation energy of X rays probably promotes ligand migration 38 and leads to self-healing of PbS CQDs³⁹ as shown in supplementary Fig. S11. To test the idea, the defect change of PbS CQDs film under X-ray irradiation were characterized by variable temperature conductance measurement (supplementary Fig. S12). The defect depth of the PbS CQDs film decreases from 0.122 eV to 0.101 eV after X-ray irradiation..... The deep understanding of this positive effect needs further investigation.” Yes on the last question, but these are nice measurements. However, why did you limit the stability study to short times (minutes or hours)... How is the stability over many days or weeks?

Response: We are thankful for the reviewer’s comment. We monitored the photoresponse of PbS CQDs film under X-ray irradiation (5.5 mGy_{air} s⁻¹) for longer times. We supplemented the stability of PbS CQDs film under X-ray irradiation in the

article as below. The photoresponse of 7 PbS CQDs films remains stable under X-ray irradiation for one week.

(Line 10-12, Page 9)

We revised the manuscript accordingly “The photoresponse of 7 PbS CQDs films remains stable after X-ray irradiation ($5.5 \text{ mGyair s}^{-1}$) for one week with a total dosage of 3326 Gyair dosage (supplementary Fig. S13).”

(Supporting Information)

We revised the supporting information file accordingly.

Fig. S13| Photoresponse of 7 PbS CQDs films under X-ray irradiation.

15. Results, pg. 9: “...and slightly decreases by 5% at bending angle of 60° possibly due to the ITO breaking.” Did you ensure that the exposed surface area is the same?

Response: Thanks the reviewer for raising this concern. We bent the CQDs photodiode at various angles and then released it to original flat state for the photoresponse tests. Hence, the exposed surface area is the same in the photoresponse tests. Through morphology characterization as shown in Fig. R5, we observed stripped cracks on the surface of the ITO film after 60° bending. We suspected that the slight degradation of device performance was due to ITO damage after 60° bending.

Fig. R5. SEM image of ITO film after 60° bending

16. Fig. S16 captions: “The weight values of X-ray image, visible image and NIR image are both 0.3.” Change “both” to “all”. (“both” implies two images instead of three).

Response: We are thankful for the reviewer’s reminding. We corrected the error in the captions.

(Supporting Information)

We revised the supporting information file accordingly “Fig. S19| Fused images with different weights coefficients. a, The weight values of X-ray image, visible image and NIR image are **all 0.3.**”

Reviewer #3 (Remarks to the Author):

The manuscript by Liu et al. presented their research on flexible detectors based on PbS colloidal quantum dots. The significant achievements here are a single detector can be used for UV, VIS, NIR and especially X-ray (thanks to Pb content in the QDs); then demonstration for multispectral image fusion with a detector array is compelling. I acknowledge the hard work and nice results of the research but I can not justify it being published in Nature Communications due to the following reasons.

1. The detector structure is standard, and performance is not superior either; many demonstrations have been demonstrated already. Very quick search, we can find PbS QD photodetectors with a responsivity of 373 A/W and a detectivity of 10^{13} Jones (Nanotechnology 32 195502) much better than the current manuscript. The X-ray response is stated as “compete well with the reported X-ray direct detectors” but the reference is from 2003. How can it compare with new results such as Nat Commun 9, 2926 (2018)? The possible significance here might be the array structure and X-ray detection with a photodetector device. But an array is just an incremental engineering demonstration, and I have no doubt that previous PbS photodetector devices in literature respond to X-rays as well.

Response: We thank the reviewer for the appreciation of the main idea of our manuscript: multispectral image fusion with a detector array is compelling compared with the existing approach using multiple detectors and complicated algorithm as shown in Fig. R6. We here answer the concerns briefly first:

1. This is the first report of pixel-level multi-spectral image fusion by one single sensor. This method could avoid pixel mismatch, overloaded computing resources and complicated systems compared with traditional methods using multiple sensors.
2. Finding a material with good response toward X-ray all the way to infrared is not easy; PbS CQDs is such a carefully chosen material.
3. The performance of our PbS CQD device toward both infrared and X-ray detection are among the best in the field.

Please read the detailed response in the following:

Multi-spectral image fusion can combine the most valuable information from different-

wavelength source images to generate a single image, which aims to be more informative and beneficial for subsequent applications, such as machine vision, autonomous vehicles, medical diagnosis and other artificial intelligences. The existing approaches for multi-spectral image fusion are mainly depended on vision algorithm including multi-scale transformation, deep learning and etc., at the sacrifice of pixel mismatch, overloaded computing resources and complicated systems. Along the breakthrough of photodetectors to broaden response band, pixel-level image fusion, capture of multi-spectral images by one single photodetector, is a more straightforward approach for simplifying image processing and system, economizing computing source and electric power

Fig. R6. a, Traditional pixel-level X-ray to infrared image fusion by using three photodetectors at corresponding wavebands and vision algorithm. b, Pixel-level X-ray to infrared image fusion by using one single flexible and broadband photodetector.

Various materials such as halide perovskites, organic semiconductors, 2D materials and colloidal quantum dots (CQDs) have emerged, enabling flexible and wide detection range beyond traditional silicon and InGaAs photodetectors as shown in Fig. R7. Halide perovskites are ultra-sensitive and have a low detection limit for X-ray and

visible detection due to their high absorption coefficient and high $\mu\tau$ product, but they show poor performance for infrared detection owing to their large bandgap. [doi.org/10.1039/C8ME00022K; doi.org/10.1126/sciadv.abg6716; doi.org/10.1038/s41928-021-00662-1] Two-dimensional materials such as graphene exhibit fast photoresponse and ultra-broadband response from visible to terahertz, but they are too thin to efficiently absorb X-ray and have limited capacities for imaging array. [doi.org/10.1126/sciadv.abf7358; doi.org/10.1002/adfm.202111970] Organic semiconductors have achieved ultra-low dark current, large linear dynamic range and excellent flexibility but with limited response range and poor X-ray irradiation resistance. [doi.org/10.1126/science.aba2624; doi.org/10.1007/s41061-021-00357-3] PbS CQDs are widely recognized for their excellent visible and infrared photodetection capabilities. Actually, PbS CQDs contain heavy element Pb which is a strong absorber for X-ray, which are at least one of the best choices for the pixel-level X-ray to infrared image fusion.

Fig. R7. Spectral ranges of various photodetectors. [1] doi.org/10.1038/nphoton.2015.280; [2] doi.org/10.1002/sml.202003397; [3] doi.org/10.1039/C8ME00022K; doi.org/10.1126/sciadv.abg6716; doi.org/10.1038/s41928-021-00662-1; [4] doi.org/10.1126/sciadv.abf7358; doi.org/10.1002/adfm.202111970; [5] doi.org/10.1126/science.aba2624; doi.org/10.1007/s41061-021-00357-3.

Despite PbS CQDs photodiodes with high responsivity have been reported in recent years, [doi.org/10.1088/1361-6528/abcc20] the prevalent problem of high dark current density persists in these devices as shown in Table R2. Furthermore, the inferred

detectivity of these devices exceeds the measured results about one to two orders of magnitude, due to the only consideration of shot noise but ignorance of generation-recombination noise and $1/f$ noise. [doi.org/10.1002/sml.202003397; doi.org/10.1021/acsnano.8b09815] In this manuscript, we achieved the lowest dark current density (12.6 nA/cm^2) by employing an all-inorganic ligand and transport layer structure, coupled with meticulous optimization of the device structure, film thickness, and other key parameters. We measured the total current noise spectrum of PbS CQD photodiodes by a lock-in amplifier, and **the corresponding measured detectivity (7.5×10^{12} Jones at 1 kHz) is the highest among the reported PbS CQDs flexible photodiodes (Fig. R8).**

Table R2. Performance list of PbS CQDs photodiodes.

Substrate	λ (nm)	EQE (%)	Dark current density (nA/cm^2)	Detectivity measured (Jones)	Detectivity inferred (Jones)	Response time (μs)	Year
Rigidity	<1100	43500 @1064	$\sim 3.9 \times 10^7$ (-1 V)	/	4.01×10^{13}	560	2021 ^[1]
Rigidity	300-1100	166 @930	~ 1000 (-1 V)	2.1×10^{11}	/	160	2014 ^[2]
Rigidity	400-1100	~ 35 @920	~ 28500 (-1 V)	/	4.91×10^{12}	~ 30	2017 ^[3]
Rigidity	400-1650	$>40\%$ @1000	~ 150 (-1 V)	1.4×10^{12}	/	/	2019 ^[4]
Rigidity	400-1300	63% @970	~ 20 (-1 V)	2.1×10^{12}	/	1.86	2022 ^[5]
Flexibility	400-1400	34% @1300	190 (-0.5 V)	6.36×10^{12}	/	8	2023 ^[6]
Flexibility	X-ray 350–1300	43.4% @970	12.6 (-0.1 V) 50.9 (-1 V)	1.01×10^{12} (1 Hz) 7.5×10^{12} (1k Hz)	/	5.1	This work

[1] doi.org/10.1088/1361-6528/abcc20; [2] doi.org/10.1002/adom.201400023; [3] doi.org/10.1039/c7ra10422g; [4] doi.org/10.1021/acsnano.9b06125; [5] doi.org/10.1038/s41928-022-00779-x; [6] doi.org/10.1016/j.scib.2023.03.016.

Fig. R8. Statistics of D^* and dark current density of PbS CQDs photodiodes.

We compare the performance of the X-ray detectors using new materials, as illustrated in Table R3. Our PbS CQDs detector can operate at the lowest bias voltage of 0~0.1 V, demonstrating a volume sensitivity of $200 \mu\text{C}\cdot\text{mGy}^{-1}\cdot\text{cm}^{-3}$ ($2\times 10^5 \mu\text{C}\cdot\text{Gy}^{-1}\cdot\text{cm}^{-3}$) to 50 keV X-ray photons, which aligns with the average performance of the reported X-ray detectors (Fig. R9). There is only one report about the X-ray detector based on PbS CQDs, which is fabricated by blending CQDs with organic materials to form a bulk heterojunction (doi.org/10.1016/j.orgel.2016.03.023). Limited by the low mobility of the blending, the volume sensitivity of this detector is merely $0.42 \mu\text{C}\cdot\text{mGy}^{-1}\cdot\text{cm}^{-3}$, significantly inferior to our device.

Table R3. The key parameters of flexible X-ray detectors.

Materials	Volume Sensitivity ($\mu\text{C mGy}^{-1} \text{cm}^{-3}$)	Bias (V)	Energy (KeV)	Year
PTAA-Bi ₂ O ₃	0.2	200	17.5	2012 ^[1]
P8T2	0.158	50	17	2009 ^[2]
TIPS-pentacene	72	0.2	17	2016 ^[3]
Cs ₂ AgBiBr ₆	4	400	45	2018 ^[4]
Bi ₂ O ₃	1712	10	50	2018 ^[5]
Ga ₂ O ₃	271	50	40	2019 ^[6]

MAPb(I _{0.9} Cl _{0.1})I ₃	362	12	60	2020 ^[7]
Cs _{0.1} (FA _{0.83} MA _{0.17}) _{0.9} Pb(Br _{0.17} I _{0.83}) ₃	55.8	0.1	70	2020 ^[8]
SCU-13	13	100	80	2020 ^[9]
Cs ₄ PbI ₆	305	10	30	2021 ^[10]
FAPbI ₃	284.2	0.5	35	2021 ^[11]
Cs ₂ TeI ₆	217	5	20	2021 ^[12]
Ni-DABDT	19.72	1	26	2021 ^[13]
DABCO-CsBr ₃	0.533	200	40	2022 ^[14]
Cs ₂ TeI ₆	1512	10	29	2022 ^[15]
MA ₃ Bi ₂ I ₆	206.5	200	30	2022 ^[16]
MPAZE-NH ₄ I ₃ ·H ₂ O	49.38	20	22	2023 ^[17]
PbS CQDs	200	0.1	50	This work

1. doi.org/10.1088/0957-4484/23/23/235502; 2. doi.org/10.1117/12.829619; 3. doi.org/10.1038/ncomms13063; 4. doi.org/10.1039/c8tc01564c; 5. doi.org/10.1038/s41467018-05301-6; 6. doi.org/10.1021/acsp Photonics.8b00769; 7. doi.org/10.1038/s41566-0200678-x; 8. doi.org/10.1021/acssami.9b14649; 9. doi.org/10.1016/j.jlumin.2021.118589; 10. doi.org/10.1021/acs.nanolett.1c03359; 11. doi.org/10.1021/acssami.0c20973; 12. doi.org/10.1021/acssami.1c04252; 13. doi.org/10.1021/acs.nanolett.1c02336; 14. doi.org/10.1021/acs.nanolett.2c02071; 15. doi.org/10.1016/j.jcis.2022.06.003; 16. doi.org/10.1002/ange.202209320; 17. doi.org/10.1002/anie.202218349;

Fig. R9. Performance comparison of flexible X-ray detectors.

In general, we demonstrated a flexible PbS CQDs photodiode array with ultra-broadband response range from X-ray to near infrared that compatibly integrates with silicon-based or flexible TFT readout circuit. Operating at an exceptionally low bias voltage (0-0.1 V), this array demonstrates outstanding performance in detecting X-ray, visible and infrared light, thus satisfying the application requirements for pixel-level multi-spectral image fusion.

(Line28-30, Page 8)

We revised the manuscript accordingly “It should be noted that the volume sensitivity of the device is about $2 \times 10^5 \mu\text{C Gy}^{-1} \text{cm}^{-3}$ at the lowest bias voltage of 0~0.1 V, which is comparable with that of the reported flexible X-ray direct detectors using new materials (supplementary Table S3)³⁴.”

(Supporting Information)

Table S3| The key parameters of flexible X-ray detectors.

Materials	Volume Sensitivity (μC mGy ⁻¹ cm ⁻³)	Bias (V)	Energy (KeV)	Year
PTAA-Bi ₂ O ₃	0.2	200	17.5	2012 ⁷
P8T2	0.158	50	17	2009 ⁸
TIPS-pentacene	72	0.2	17	2016 ⁹

Cs₂AgBiBr₆	4	400	45	2018¹⁰
Bi₂O₃	1712	10	50	2018¹¹
Ga₂O₃	271	50	40	2019¹²
MAPb(I_{0.9}Cl_{0.1})I₃	362	12	60	2020¹³
Cs_{0.1}(FA_{0.83}MA_{0.17})_{0.9} Pb(Br_{0.17}I_{0.83})₃	55.8	0.1	70	2020¹⁴
SCU-13	13	100	80	2020¹⁵
Cs₄PbI₆	305	10	30	2021¹⁶
FAPbI₃	284.2	0.5	35	2021¹⁷
Cs₂TeI₆	217	5	20	2021¹⁸
Ni-DABDT	19.72	1	26	2021¹⁹
DABCO-CsBr₃	0.533	200	40	2022²⁰
Cs₂TeI₆	1512	10	29	2022²¹
MA₃Bi₂I₆	206.5	200	30	2022²²
MPAZE-NH₄I₃·H₂O	49.38	20	22	2023²³
PbS CQDs	200	0.1	50	This work

2. 900 nm thickness of PbS is stated to be determined by the diffusion and drift length of photogenerated carriers and adequate X-ray absorption. This statement is very standard, all researchers know such information but how to get 900nm is a mystery. Is it really optimized or simply a one-shot?

Response: Thanks the reviewer for raising this concern. 900 nm is a balanced thickness for our detector considering its NIR and X-ray detection performance. We made PbS CQDs photodiodes with different thickness of CQDs layer and added their photoresponses as supplementary Fig. S5. Thicker CQDs layer enhances X-ray and NIR absorption. The high penetration depth of X-ray enables photogenerated carriers within or near the depleted region, which facilitates effective extraction of photogenerated

carriers. The photoresponse to X-ray is enhanced by increasing the thickness of CQDs layer. However, for NIR illumination, the photogenerated carriers are mainly at the surface of CQDs layer far from the depletion region, resulting in low extraction efficiency and hence lower performance. Considering the contradictory requirement, 900 nm is the optimal thickness for our device.

(Supporting Information)

Fig. S5| Photoresponse of PbS CQDs photodiodes with different thickness of PbS CQDs layer. a, Transient responses at -0.1 V bias under $5.1 \text{ mGy}_{\text{air}} \text{ s}^{-1}$ dose rates X-ray. **b,** Transient responses at -0.1 V bias illuminated by 970 nm LED with a power density of 0.45 mW cm^{-2} . Photogenerated carrier transmission in CQDs photodiodes under **(c)** X-ray and **(d)** Vis-NIR irradiation³.

3. Basically, I did not learn much new knowledge from this manuscript rather than seeing a fancy demonstration, which is worth publishing but in a specialized journal.

Response: We appreciate the reviewer for the valuable remarks. In this work, we demonstrated a flexible PbS CQDs photodiode array with ultra-broadband response range from X-ray to near infrared, which has impressive performance with a low dark

current density, a high detectivity under visible-near infrared illumination and a comparable sensitivity under X-ray irradiation. The main innovations of this work are as follows.

1. We demonstrated a simple method for pixel-level multi-spectral image fusion by one single sensor for the first time, avoiding pixel mismatch, overloaded computing resources and complicated systems compared with traditional methods. This new approach could be useful for flexible lens-free imaging, such as biomedical measurement and medical diagnosis.
2. This work systematically showed flexible and broadband PbS CQDs photodiode array for pixel-level image fusion from X-ray to near-infrared. This array achieves the lowest dark current (12.6 nA/cm^2) and the highest measured detectivity (7.5×10^{12} Jones at 1 kHz) among the reported PbS CQDs flexible photodiodes in both visible light and infrared bands. It also exhibits an impressive X-ray sensitivity and operates at an exceptionally low bias voltage (0-0.1 V), thus satisfying the application requirements for multi-spectral image fusion.
3. This study reported for the first time the positive effect of X-ray irradiation on the device performance of PbS CQD devices and presented a preliminary explanation for this observation. For polycrystalline PbS film, as in other bulk materials, atomic displacement and hence lattice defects (Fig. R10a) would be generated by the ionizing radiation, resulting in deteriorated device performance [doi.org/10.1016/j.jnucmat.2018.10.027; doi.org/10.1021/acsaelm.0c00854; doi.org/10.1002/sml.202107516]. PbS CQDs are of large specific surface area and quasi-amorphous, of which the surface exists many unsaturated bonds and vacancies (Fig. R10b). The irradiation energy of X-ray photons probably promotes ligand migration and defect annihilation, and therefore leads to enhanced device performance.

Fig. R10. a, Schematic illustration of the damage on PbS crystal by X-ray irradiation.
 b, Schematic diagram of the ligand migration on PbS CQDs under X-ray irradiation.

REVIEWERS' COMMENTS

Reviewer #1 (Remarks to the Author):

All my comments have been fully addressed by the authors, the manuscript is ready for publication in this journal.

Reviewer #2 (Remarks to the Author):

The authors added substantial technical information to effectively address all of my comments/questions and the information presented in the manuscript is much stronger.

Reviewer #3 (Remarks to the Author):

The authors revised the manuscript to address the raised concerns by all reviewers. Authors tried to convince that this is the first time of pixel-level multi-spectral image fusion by one single sensor, which was well understood by all reviewers. We all know that the device is not new (though the array is new). The performance is also not superior if we do comparison with literature for individual parameters. Combination of all the factors, this device might have some advantages. I do not see the authors consider the comparison with Nat Commun 9, 2926 (2018) as I suggested.

One more thing that authors highlighted certainly understand as the most significance of the work is the multi-spectral image fusion by one single sensor. However, to achieve this we need to scarify the image quality of visible and NIR imaging because of no lens and large pixel size. It is hard to imagine to do optical imaging only at the X-ray imaging resolution. While the computational imaging techniques and simple calibration of the three sepearated sensors can be done pretty easily.

Response Letter

Reviewer #1 (Remarks to the Author):

All my comments have been fully addressed by the authors, the manuscript is ready for publication in this journal.

Response: We appreciate the reviewer for the valuable and positive feedback of our work.

Reviewer #2 (Remarks to the Author):

The authors added substantial technical information to effectively address all of my comments/questions and the information presented in the manuscript is much stronger.

Response: We are grateful that the reviewer appreciate our work.

Reviewer #3 (Remarks to the Author):

The authors revised the manuscript to address the raised concerns by all reviewers. Authors tried to convince that this is the first time of pixel-level multi-spectral image fusion by one single sensor, which was well understood by all reviewers. We all know that the device is not new (though the array is new). The performance is also not superior if we do comparison with literature for individual parameters. Combination of all the factors, this device might have some advantages. I do not see the authors consider the comparison with Nat Commun 9, 2926 (2018) as I suggested.

One more thing that authors highlighted certainly understand as the most significance of the work is the multi-spectral image fusion by one single sensor. However, to achieve this we need to scarify the image quality of visible and NIR imaging because of no lens and large pixel size. It is hard to imagine to do optical imaging only at the X-ray imaging resolution. While the computational imaging techniques and simple calibration of the three seperated sensors can be done pretty easily.

Response: We thank the reviewer for the appreciation of pixel-level multi-spectral image fusion in our work. Despite PbS CQDs photodiodes have been reported in recent years, the ultra-broadband flexible imaging array remains unexplored. In this

manuscript, our device obtained the lowest dark current density (12.6 nA/cm^2) and the highest measured detectivity (7.5×10^{12} Jones at 1 kHz) among the reported PbS CQDs flexible photodiode. The device could operate at the lowest bias voltage of 0~0.1 V, demonstrating a volume sensitivity of $200 \text{ } \mu\text{C} \cdot \text{mGy}^{-1} \cdot \text{cm}^{-3}$ to 50 keV X-ray photons, which aligns with the average performance of the reported flexible X-ray detectors.

We have added a comparison with Nat. Commun. 9, 2926 (2018) in the last-version supporting information. The mentioned reference introduced a flexible direct X-ray detector, which was fabricated by incorporating Bi_2O_3 nanoparticles into an organic bulk heterojunction in Nat. Commun. 9, 2926 (2018). Though the Bi_2O_3 X-ray detector has a volume sensitivity of $1712 \text{ } \mu\text{C} \cdot \text{mGy}^{-1} \cdot \text{cm}^{-3}$ at bias of -10 V , our device exhibits higher sensitivity at lower operating biases ($200 \text{ } \mu\text{C} \cdot \text{mGy}^{-1} \cdot \text{cm}^{-3}$ at 0.1 V) comparing with $0.13 \text{ } \mu\text{C} \cdot \text{mGy}^{-1} \cdot \text{cm}^{-3}$ at -2.5 V of the Bi_2O_3 X-ray detector. The low working bias supports better suitability for portable monitoring. Furthermore, Bi_2O_3 nanoparticles have a band gap of 3.34 eV (doi.org/10.1088/1757-899X/763/1/012036), which can't achieve visible and infrared detection.

The multi-spectral image fusion by one single sensor is an alternative way for X-ray to infrared image fusion. For close-range imaging such as biomedical measurement and medical diagnosis, the lensless infrared imaging system demonstrates significant application potential (doi.org/10.1038/s41928-019-0354-7). Considering short wavelength and weak volatility of X-ray, only the lensless imaging system can be used for X-ray imaging. Hence, the lensless imaging system is more favorable for the multi-spectral image fusion from X-ray to infrared than the imaging system with lens. The multi-spectral image fusion by one single sensor is a better method than that by separated sensors from X-ray to infrared, which avoids pixel mismatch and saves energy and cost.